# RNA-Seq and Electrical Penetration Graph Revealed the Role of *Grh1*-Mediated Activation of Defense Mechanisms towards Green Rice Leafhopper (*Nephotettix cincticeps* Uhler) Resistance in Rice (*Oryza sativa* L.)

**DOI:** 10.3390/ijms221910696

**Published:** 2021-10-02

**Authors:** Youngho Kwon, Nkulu Rolly Kabange, Ji-Yoon Lee, Bo Yoon Seo, Dongjin Shin, So-Myeong Lee, Jin-Kyung Cha, Jun-Hyeon Cho, Ju-Won Kang, Dong-Soo Park, Jong-Min Ko, Jong-Hee Lee

**Affiliations:** 1Department of Southern Area Crop Science, National Institute of Crop Science, RDA, Miryang 50424, Korea; kwon6344@korea.kr (Y.K.); minitia@korea.kr (J.-Y.L.); jacob1223@korea.kr (D.S.); olivetti90@korea.kr (S.-M.L.); jknzz5@korea.kr (J.-K.C.); hy4779@korea.kr (J.-H.C.); kangjw81@korea.kr (J.-W.K.); parkds9709@korea.kr (D.-S.P.); kojmin@korea.kr (J.-M.K.); 2Crop Protection Division, National Institute of Agricultural Science, RDA, Jeonju 55365, Korea; seoby@korea.kr

**Keywords:** RNA-Seq, transcriptome profiling, regulatory pathways, green rice leafhopper, insect pest–plant interaction, rice

## Abstract

The green rice leafhopper (GRH, *Nephotettix cincticeps* Uhler) is one of the most important insect pests causing serious damage to rice production and yield loss in East Asia. Prior to performing RNA-Seq analysis, we conducted an electrical penetration graph (EPG) test to investigate the feeding behavior of GRH on Ilpum (recurrent parent, GRH-susceptible cultivar), a near-isogenic line (NIL carrying *Grh1*) compared to the *Grh1* donor parent (Shingwang). Then, we conducted a transcriptome-wide analysis of GRH-responsive genes in Ilpum and NIL, which was followed by the validation of RNA-Seq data by qPCR. On the one hand, EPG results showed differential feeding behaviors of GRH between Ilpum and NIL. The phloem-like feeding pattern was detected in Ilpum, whereas the EPG test indicated a xylem-like feeding habit of GRH on NIL. In addition, we observed a high death rate of GRH on NIL (92%) compared to Ilpum (28%) 72 h post infestation, attributed to GRH failure to suck the phloem sap of NIL. On the other hand, RNA-Seq data revealed that Ilpum and NIL GRH-treated plants generated 1,766,347 and 3,676,765 counts per million mapped (CPM) reads, respectively. The alignment of reads indicated that more than 75% of reads were mapped to the reference genome, and 8859 genes and 15,815,400 transcripts were obtained. Of this number, 3424 differentially expressed genes (DEGs, 1605 upregulated in Ilpum and downregulated in NIL; 1819 genes upregulated in NIL and downregulated in Ilpum) were identified. According to the quantile normalization of the fragments per kilobase of transcript per million mapped reads (FPKM) values, followed by the Student’s *t-*test (*p* < 0.05), we identified 3283 DEGs in Ilpum (1935 upregulated and 1348 downregulated) and 2599 DEGs in NIL (1621 upregulated and 978 downregulated) with at least a log_2_ (logarithm base 2) twofold change (Log_2_FC ≥2) in the expression level upon GRH infestation. Upregulated genes in NIL exceeded by 13.3% those recorded in Ilpum. The majority of genes associated with the metabolism of carbohydrates, amino acids, lipids, nucleotides, the activity of coenzymes, the action of phytohormones, protein modification, homeostasis, the transport of solutes, and the uptake of nutrients, among others, were abundantly upregulated in NIL (carrying *Grh1*). However, a high number of upregulated genes involved in photosynthesis, cellular respiration, secondary metabolism, redox homeostasis, protein biosynthesis, protein translocation, and external stimuli response related genes were found in Ilpum. Therefore, all data suggest that *Grh1*-mediated resistance against GRH in rice would involve a transcriptome-wide reprogramming, resulting in the activation of bZIP, MYB, NAC, bHLH, WRKY, and GRAS transcription factors, coupled with the induction of the pathogen-pattern triggered immunity (PTI), systemic acquired resistance (SAR), symbiotic signaling pathway, and the activation of genes associated with the response mechanisms against viruses. This comprehensive transcriptome profile of GRH-responsive genes gives new insights into the molecular response mechanisms underlying GRH (insect pest)–rice (plant) interaction.

## 1. Introduction

During their life cycle, plants are constantly confronted by different sorts of environmental stimuli (abiotic stress) [1,2] or attacks from living organisms (biotic stress), including pathogens such as bacteria, viruses, fungi, nematodes, and insect pests causing serious damages to crops. Despite their fundamental roles in food production [3,4], insects are said to be responsible for a reduction in the yield of crops, which results in economic losses across the world [5,6], thus putting at risk the food security. Various measures have been proposed to control and minimize these losses, and the most practical and economical control measure is the use of resistant varieties to insect pests. 

Rice (*Oryza sativa* L.) is recognized as the staple food for nearly half of the global population, which is projected to reach about 9.8 billion people by 2050 [7]. The rapid increase in the global population has been the key driver to increasing rice production and productivity across the globe. This cereal crop (rice), solely cultivated for human consumption, is the host plant to many insect pests, including planthoppers (brown planthopper (BPH): *Nilaparvata lugens* Stal, small brown planthopper (SBPH): *Laodelphax striatellus* Fallen, green leafhopper (GLH): *Nephotettix virescens* Distant, green rice leafhopper (GRH): *Nephotettix cincticeps* Uhler, whitebacked planthopper (WBPH): *Sogatella furcifera* Horvath, and zigzag leafhopper (ZLH): *Recilia dorsalis* Motschulsky), stem borers, and gall midges, which are considered as the most serious pests of rice [8]. Among them, GRH, known as one of the most important insect pests threatening rice production in East Asia [9], has been identified as a vector for the rice dwarf virus (RDV), rice waika virus (RWV) [10], rice transitory yellow virus (RTYV) [11], and rice yellow dwarf virus (RYDV). The control of insect pests such as GRH in rice production is a challenging task, involving the use of pesticides or a biological control approach. These methods are costly and harmful to the environment. The use of resistant rice lines has been proposed as the most effective way to overcome crop failure caused by GRH and viral diseases conveyed by this insect pest [11,12].

Intrinsic to their mode of action, GRHs suck sap from both the xylem and the phloem of susceptible rice varieties, leading to plant growth failure [13] and important economic losses [14]. According to Matsumoto et al. [15], during the feeding process on the rice plant, the GRH accumulates bioactive proteins such as laccase and beta-glucosidase in its salivary glands [16,17], which have the ability to hinder the defense system of the host plant, eventually enabling the insect to ingest nutrients derived from the host, resulting in cell death [18,19].

Upon insect pest attack, plants activate the appropriate defense mechanism, tending to provide a proper level of resistance to combat the stress. During this event, defense-related genes are induced, while antioxidant systems (nonenzymatic and enzymatic) and phytohormone signaling cascades are activated [20]. Among the plant hormones involved in the plant stress response mechanism against biotic stresses, salicylic acid (SA) and jasmonic acid (JA) have been shown to be predominant [21]. Under the same conditions, a transcriptional reprogramming within the cell takes place, which is followed by the accumulation of protein kinases, secondary metabolites, and changes in the photosynthetic process, among other cellular processes [22].

For several decades, many plant breeding research programs have shown a growing interest in developing crop varieties with improved resistance against economically important insect pests, including GRH. In recent years, the use of molecular breeding techniques such as marker-assisted selection (MAS) or marker-assisted backcrossing (MABC), as well as gene pyramiding on various types of breeding populations, such as near-isogenic lines (NILs), recombinant inbred lines (RILs), doubled haploid (DH) lines, and pyramiding lines, have proven beneficial for developing rice varieties carrying a single gene or multiple GRH resistance genes [23,24].

So far, studies aimed at investigating quantitative trait loci (QTLs) associated with GRH resistance in rice have employed forward genetics, in addition to linkage mapping and QTL analysis or genome-wide association studies (GWAS) approaches. These studies have proposed a number of QTLs mapped to different chromosomes of rice, and, in some cases, fine mapping of major QTLs has been performed, resulting in the identification of putative candidate genes (*Grh1*, chromosome 5 [25]; *Grh2,* chromosome 11; *Grh3*, chromosome 6; *Grh4,* chromosome 3; *Grh5,* chromosome 8 [13]; *Grh6,* chromosome 4; *Grh9* [23,26]). Of this number, a few genes have been cloned and functionally characterized.

Recent advances in molecular breeding research and the emergence of genome sequencing technologies have introduced a wide range of opportunities and possibilities for exploring the molecular basis underlying the defense mechanisms of plants in response to pathogens or attack of pests. As sequencing technology has improved and costs have decreased, an exponential increase in the use of transcriptome studies has been observed, and RNA sequencing (herein referred to as RNA-Seq) has been employed to explore the molecular mechanisms of various biological phenotypes [27,28,29,30,31,32]. In addition to the conventional use of gene annotation, profiling, and expression comparison, transcriptome studies have been applied for many other purposes, such as gene structure analysis, identification of novel genes or regulatory RNAs, RNA editing analysis, co-expression or regulatory network analysis, biomarker discovery, development-associated imprinting studies, single-cell RNA sequencing studies, and pathogen–host dual RNA sequencing studies. 

This study aimed to investigate the transcriptome-wide profile, as well as the defense mechanisms and signaling pathways activated, when rice plants are infested with green rice leafhopper (GRH). To achieve that, we used a rice near-isogenic line (NIL) reported by Park et al. [33] (carrying the *Green rice leafhopper resistance 1* (*Grh1)* locus, which has been fine mapped to chromosome 5 and located in a region covered by 670 kbp) and the rice cultivar Ilpum (GRH-susceptible) exposed to GRH infestation. Then, we performed a transcriptome study to identify novel GRH-responsive genes between the NIL GRH-resistant line and Ilpum. In addition, we monitored the feeding behavior of GRH, over time, on GRH-resistant and -susceptible rice lines, using the electrical penetration graph (EPG) technique to investigate the molecular mechanisms underlying *Grh1*-mediated resistance against GRH, and we elucidated the molecular basis for the GRH (insect pest)–rice (plant) interaction, which would explain the differential transcriptome profile between GRH-susceptible and -resistant rice lines, as well as the activation or suppression of diverse regulatory pathways, coupled with reprogramming of physiological and biochemical processes. 

## 2. Results

### 2.1. Feeding Behavior of GRH Reared on Ilpum and NIL Carrying Grh1

Electrical penetration graph (EPG) technology has long been employed to observing the stylet probing behavior of waveforms in each process, from host detection to insect feeding, and it helps examine the phloem feeding activity of the insect on the host plant [34]. Waveforms, classified from Nc1 to Nc6, are shown for each feeding pattern. The Nc6 waveform implies that the GRH sucks the phloem sap [35]. The EPG results in panels A and B of Figure 1 show the feeding behavior of GRH on Ilpum and NIL, respectively. These results indicate that GRH could not penetrate the stylet in the phloem of NIL (as indicated by the Nc5 waveform) compared to Ilpum. This feeding pattern was similarly observed when GRH were reared on Shingwang (the donor parent of *Grh1*) showing zero (0) in the cumulative Nc6 column (see Appendix A). The accumulated Nc6 waveform in Ilpum implies that GRH had took up substances from the phloem, as indicated by the Nc6 average value of 137.4 ± 126.1 min.

In addition to the feeding habit of GRH on the susceptible and resistant rice lines (Figure 1), we estimated the survival percentage of GRH reared on Ilpum and NIL. Results indicate that GRH reared on Ilpum (GRH susceptible) recorded a significant and sustained survival percentage over time, with 72% of insects still alive 72 h after infestation. In contrast, 60% of insects died 36 h after being infested in NIL (near-isogenic line) seedlings carrying the *Grh1* resistance locus. The death rate increased with time and reached 92% (or 8% of alive GRH) at the final count (72 h after infestation) (Figure 2).

### 2.2. Transcriptome Analysis

To investigate the changes in gene expression profile during GRH feeding-mediated biotic stress, rice seedlings placed in test tubes were infested with 10 insects (GRH) per tube, and leaf samples were collected 48 h post infestation, in duplicate, for RNA-Seq analysis of Ilpum (recurrent parent) and NIL. Another set of leaf samples were collected from seedlings grown without GRH. Data in Figure 3 indicate highly significant changes in gene expression profile at *p* < 0.05. We observed a 13.3% increase in the number of upregulated genes in the NIL compared to Ilpum (Figure 3C,D). In the same way, the number of read counts (CPM) of Ilpum GRH-treated and NIL GRH-treated plants was 526,030 and 690,017, respectively (Figure 3A,B), which revealed about a 31.2% increase in NIL.

On the basis of the quantile normalization of the fragments per kilobase of transcript per million mapped reads (FPKM) values (followed by the Student’s *t-*test, *p* < 0.05), we identified 3283 DEGs in Ilpum (1935 upregulated and 1348 downregulated) and 2599 DEGs in NIL (1621 upregulated and 978 downregulated) with at least log_2_ (logarithm base 2) twofold change (Log_2_FC ≥2) in the expression, in response to GRH. Up to 75% of these reads successfully mapped to the rice reference genome database (Figure 3E). In total, 8859 genes and 15,815,400 transcripts were identified (Figure 3C). Of this number, 4265 and 4594 genes were upregulated and downregulated, respectively, in Ilpum vs. 4479 and 4380 up- and downregulated, respectively, in NIL. Among them, 3424 were differentially expressed genes (DEGs, 1605 upregulated in Ilpum but downregulated in NIL; 1819 genes upregulated in NIL but downregulated in Ilpum), which represents about 38.6% of the total number of expressed genes. Under the same conditions, 2660 genes were commonly upregulated and 2775 were downregulated in Ilpum and NIL, respectively.

The major characteristic of near-isogenic lines (NILs), also known as inbred lines, is the identical genetic makeups with their recurrent parent except for a few specific locations or genetic loci as a result of an introgression of a desirable character or trait (gene) from a donor parent into an otherwise agronomically acceptable cultivar (recurrent parent) [36]. Here, Ilpum and NIL had the same genetic background, except the genetic locus covering *Grh1*, a GRH-resistance locus introduced from the rice cultivar Shingwang (donor parent) to Ilpum (recurrent parent). This is reflected in the heat map (Figure 3D), which shows the pattern of DEGs and commonly expressed genes in response to GRH. The quantile normalization of the fragment per kilobase of transcript per million mapped reads (FPKM) values (value 1 and value 2) and the selection of DEGs having at least twofold change (Log_2_FC) in their transcriptional level in response to GRH infestation led to 2250 DEGs being identified in Ilpum, of which 1410 genes were upregulated and 840 were downregulated. Similarly, 1834 DEGs were identified (1237 upregulated and 597 downregulated genes). The gene expression and raw sequence data were submitted to the Gene Expression Omnibus (GEO) and Short Read Archive (SRA) at NCBI (https://www.ncbi.nlm.nih.gov/geo/submitter/), accessed on 25 August 2021, with accession numbers GSE176497 and SRP323419, respectively, using FileZilla 3.54.1 software (Tim Kosse, FileZilla ©2004-20021, https://filezilla-project.org/), accessed on 25 August 2021. Lists of the 10 most upregulated and 10 most downregulated genes in Ilpum and NIL are given in Appendix A and Appendix A, respectively, and discussed below in detail.

Because a large number of genes, classified in diverse regulatory pathways, showed differential transcriptome profiles between Ilpum and NIL, we were interested to unveil the identity of DEGs that exhibited opposite expression patterns. The results of the comparative transcriptome profiling indicate that, of the top 20 DEGs, 10 were upregulated in Ilpum by GRH (7.6–12.6 log_2_FC) but downregulated in NIL (–1.6 to –9.9 log_2_FC) (Table 1). Meanwhile, the other 10 DEGs showed an opposite transcriptional pattern (downregulated in Ilpum by –0.13 to –8.3, while being upregulated in NIL by 8.4–12.5 log_2_FC). In the same way, the top 10 upregulated genes showed an increase in their expression levels by 8.1–15.4 log_2_FC in NIL, while having lower expression levels in Ilpum under the same conditions (upregulated by 0.2–2.7 log2FC) (Table 2).

### 2.3. Functional Classification of Genes: Gene Ontology Enrichment Analysis

In the perspective of identifying the classes to which the identified GRH-responsive genes belong, a Gene Ontology (GO) classification was done, according to the molecular function, biological process, and cellular response. The GO enrichment revealed that a broad range of molecular functions and biological processes were involved, among which the binding activity, transport activity, and catalytic activity predominated. The majority of the identified genes were proposed to be involved in the macromolecule catabolic process, organic acid biosynthetic process, carboxylic acid biosynthesis, protein catabolic process, or cofactor metabolism (Appendix A). In the same way, the Kyoto Encyclopedia of Genes and Genomes (KEGG) pathway analysis showed that most of the identified GRH-responsive genes belong to phytohormone biosynthesis pathways, terpenoid and steroid synthesis, alkaloid biosynthesis pathway from shikimate and terpenoids, or carbone fixation during photosynthesis (Appendix A). In addition, most of the genes encoding proteins were localized in the cell membrane, ribonucleoprotein complex, or organelle membrane (Appendix A).

### 2.4. Differentially Expressed Genes Associated with Abiotic and Biotic Stresses Response

Fifty-eight DEGs were proposed to be associated with the response to external stimuli. Of this number, 34 were associated with abiotic stress response, and 24 were associated with biotic stress response (Figure 4G). The set of DEGs associated with abiotic stress included drought stress (stomatal closure signaling: two), cold stress (cold response sensor: three), heat stress (heat response transcriptional regulator: two, temperature sensing and signaling: one), salinity (salt overly sensitive (SOS) signaling pathway: four, GA signaling pathway crosstalk: two), light (red or far red light: four, UV-A/blue light: 12), carbon dioxide (CO_2_ sensing and signaling: two), and gravity sensing and signaling (four). Among the 34 genes, 18 were differentially regulated between Ilpum and NIL (seven genes were downregulated in Ilpum (log_2_FC: −0.12 to −1.24) but upregulated in NIL (Log_2_FC: 0.03–1.46), while six other genes were upregulated in Ilpum (Log_2_FC: 0.04–2.53) but downregulated in NIL (Log_2_FC: −0.1 to −4.87)). Some of the DEGs (downregulated in Ilpum and upregulated in NIL), which are associated with the abiotic stress response, include *EID1* (a regulator component encoding the CUL4–DDB1 ubiquitination complex), *PIF* transcription factor, *PKS* (phototropin signaling factor), *ARG1* (signaling protein factor), *SCaBP8*/*CBL10* (SOS3-SOS2, salt overly sensitive signaling and calcium-dependent regulatory protein), and *DELLA* transcription factor (involved in GA signaling pathway crosstalk).

GRH infestation is primarily considered as a biotic stress, because the stress is induced by a living organism (insect pest), which is also a vector for various plant’s viruses. Here, DEGs associated with biotic stress response included the damage-elicitor peptide precursor (three), pathogen-pattern triggered immunity (PTI) network (eight), pathogen defense mechanisms (systemic acquired resistance, SAR: six), symbiotic signaling pathway (two), and virus response mechanism (five). It was interesting to see that five genes associated with pathogen response mechanism exhibited a differential transcriptional regulation between Ilpum and NIL (upregulated in Ilpum (Log_2_FC: 0.6–2.5) but downregulated in NIL (Log_2_FC: −0.09 to −0.5) (Appendix A). These genes included *ARL8* (encoding small GTPase associated with tobamovirus multiplication), *PEPR* (pep-elicitor peptide receptor involved in damage repair), and three genes (*XLG*, *AGG1/2*, and *PCRK*) associated with the pattern-triggered immunity (PTI) network and bacterial elicitor response. Furthermore, as shown in Figure 3 and Figure 4, RNA-Seq results showed differential transcriptional regulation between specific genes associated with different metabolic pathways and the cellular response to abiotic and biotic stress response between Ilpum and NIL following GRH infestation. 

### 2.5. Differentially Expressed Genes Involved in Hormone Metabolism and Other Signaling Cascades

Upon stress induction, from living organisms (biotic stress) or environmental factors (abiotic stress), plants activate various signaling cascades that are sent as an alert to activate the required defense system. In plants, phytohormones, in addition to being essential for growth and development of plants, serve as signaling molecules under stress. After being perceived, signals are transduced at a whole-plant level and allow the activation of various antioxidant systems, followed by a transcriptional reprogramming within the cell. Here, upon GRH infestation, the RNA-Seq results revealed that 168 DEGs associated with phytohormone biosynthesis or signaling (abscisic acid (ABA) biosynthesis: nine, perception and signaling: 11, and conjugation and degradation: two; auxin biosynthesis: three, perception and signal transduction: five, conjugation and degradation: two, and transport: five; brassinosteroid (BR) biosynthesis: four, and perception and signal transduction: 17; cytokinin (CK) biosynthesis: three, perception and signal transduction: 15, and conjugation and degradation: five; ethylene (ET) biosynthesis: six, and perception and signal transduction: 16; gibberellin (GA) biosynthesis: two, perception and signal transduction: one, modification and degradation: two, and transport: one; jasmonic acid (JA) biosynthesis: 15, and perception and signal transduction: eight; salicylic acid (SA) perception and signal transduction: one; strigolactone (SL) biosynthesis: one, and perception and signal transduction: eight) were induced or suppressed. Another group of 25 DEGs were categorized as signaling peptides (Figure 4F).

We were interested to see the differential transcriptional level of phytohormone signaling pathway genes in response to GRH. As shown in Figure 5 and Appendix A, of the 168 identified DEGs, 56 genes recorded differential expression patterns between Ilpum and NIL, of which 37 showed a downregulation pattern in Ilpum (Log_2_FC: −0.01 to −7.3) but were upregulated in NIL (Log_2_FC: 0.03–8.1), whereas 19 genes were upregulated in Ilpum (Log_2_FC: 0.24–3.5) but downregulated in NIL (Log_2_FC: −0.01 to −4.9) under the same conditions. However, 112 had similar expression patterns in Ilpum and NIL. The DEGs included auxin (biosynthesis, transport, conjugation and degradation, perception and signal transduction), brassinosteroids (BR biosynthesis, perception and signal transduction), cytokinin (CK biosynthesis, perception and signal transduction, conjugation and degradation), ethylene (ET biosynthesis, perception and signal transduction), gibberellin (GA biosynthesis, perception and signal transduction, modification and degradation, and transport), jasmonic acid (JA biosynthesis, perception and signal transduction), salicylic acid (SA perception and signal transduction), strigolactone (SL biosynthesis, perception and signal transduction), and other signaling peptide-related genes.

### 2.6. DEGs Associated with Secondary Metabolism and Redox Homeostasis

Secondary metabolites, also known as secondary (specialized) or natural products, are organic compounds derived from primary metabolites generated by living organisms, including plants, which are not required for the growth, development, or reproduction of the organism but are produced to confer a selective advantage to the organism. For instance, they may be involved in plants defense or act as regulators in various plant metabolic processes, among other factors [37]. In this study, in addition to the defense-related genes mentioned in the above paragraphs, the MapMan analysis of the RNA-Seq data indicated that, of all the DEGs identified following GRH infestation, 64 were associated with secondary metabolism in plants (terpenoids: 36, phenolics: 20, glucosinolates: five, betaines: two, and alkaloids: one) (Figure 4D). In addition, panel E of Figure 4 indicates that 54 DEGs were associated with reduction–oxidation (redox) homeostasis (reactive oxygen species (ROS) generation: seven, enzymatic ROS scavengers: six, low-molecular-weight scavengers: 11, hydrogen peroxide (H_2_O_2_) removal: 13, chloroplast redox homeostasis: eight, and cytosol/mitochondrion/nucleus redox homeostasis: nine).

The transcriptional regulation of genes associated with secondary metabolism was as follows: 24 genes exhibited differential transcript accumulation patterns between Ilpum and NIL, of which 12 genes were downregulated in Ilpum (Log_2_FC: −0.02 to −3.78) but upregulated in NIL, and the other 12 were upregulated in Ilpum (Log_2_FC: 0.04–1.6), while being downregulated in NIL (Log_2_FC: −0.2 to −2.7). However, the remaining 40 genes had similar expression patterns between Ilpum and NIL. Regarding the transcriptional regulation of genes associated with redox homeostasis, 27 were differentially expressed between Ilpum and NIL, where 12 genes were downregulated in Ilpum (Log_2_FC: −0.4 to −4.9) but upregulated in NIL (Log_2_FC: 0.01–1.6). Under the same conditions, 15 genes exhibited an upregulation pattern in Ilpum (Log_2_FC: 0.02–5.04), while being downregulated in NIL (Log_2_FC: −0.04 to −2.1) (Figure 5 and Figure 6, Appendix A).

### 2.7. Differentially Expressed Genes with GO Terms Involved in Cellular Functions 

MapMan analysis showed that 170 DEGs are involved in photosynthesis (Calvin cycle: 32, CAM/C4 photosynthesis: seven, photophosphorylation: 110, photorespiration: 21) (Figure 4A). Among them, 52 genes were differentially regulated. In essence, 12 were downregulated in Ilpum (Log_2_FC −0.286 to −2.372) but upregulated in NIL (Log_2_FC 0.08–6.67), whereas 40 were upregulated in Ilpum (Log_2_FC: 0.002–6.69) but downregulated in NIL (Log_2_FC: −0.21 to −11.3). In the same way, 95 and 122 DEGs were associated with cellular respiration (glycolysis: 26, oxidative phosphorylation: 44, pyruvate oxidation: eight, tricarboxylic acid (TCA) cycle: 17) and carbohydrate metabolism (fermentation: three, galactose metabolism: one, gluconeogenesis: two, mannose metabolism: four, nucleotide sugar biosynthesis: 34, oligosaccharide metabolism: five, oxidative pentose phosphate pathway: 12, plastidial glycolysis: eight, sorbitol metabolism: two, starch metabolism: 20, sucrose metabolism: 26, and trehalose metabolism: five), respectively. A comparative transcriptome analysis revealed that, among the genes controlling cellular respiration, 34 were differentially regulated (10 downregulated in Ilpum (Log_2_FC: −0.71 to −5.75) but upregulated in NIL (Log_2_FC: 0.48–5.17), and 24 upregulated in Ilpum (Log_2_FC: 0.099–4.49) but downregulated in NIL (Log_2_FC: −0.15 to −4.19)), and 61 genes showed a similar expression pattern (upregulated or downregulated) in both Ilpum and NIL. Likewise, out of the 122 DEGs involved in the carbohydrate metabolism, 55 genes were differentially regulated (31 downregulated in Ilpum (Log_2_FC: −0.06 to −13.71) but upregulated in NIL (Log_2_FC: 0.16–11.54), and 24 genes were upregulated in Ilpum (Log_2_FC: 0.28–4.36) but downregulated in NIL (Log_2_FC: −0.03 to −3.99)), while 75 genes had similar transcript accumulation patterns in both Ilpum and NIL (Appendix A).

Another set of 80 DEGs, identified as GRH-responsive genes, based on their annotation, were associated with multi-process signaling (SnRK1-kinase regulatory system: 13, programmed cell death (PCD: 11), circadian clock system: 14, organelle machinery: 12, target of rapamycin (TOR) signaling: eight, G-protein signaling: one, phosphoinositide lipid regulatory system: 27, and Rop-GTPase regulatory system: five) (Appendix A), protein modification (acetylation: five, glycosylation: 30, protein folding: 24, disulfide bond formation: 4, targeting peptide maturation: 26, hydroxylation: 3, *S-*glutathionylation: 26, *S*-nitrosylation: one, lipidation: 12, and phosphorylation: 365) (Appendix A), cell-cycle organization (cell-cycle control: 16, DNA replication: seven, mitosis and meiosis: 18, cytokinesis: 12, organelle DNA replication: three, organelle division: five, and membrane organization: 11) (Appendix A). In the same way, data identified DEGs associated with nutrient uptake (copper uptake: three, iron uptake: 14, phosphorus assimilation: 10, sulfur assimilation: four, nitrogen assimilation: 23) (Appendix A) and protein homeostasis (ubiquitin fold protein conjugation: 117, ubiquitin-proteasome system: 49, Cullin: 46, Hsp90 chaperone system: 11, cytosolic Hsp70 chaperone system: 17, plastidial Hsp70 chaperone system: five, ribosome-associated chaperon activities: eight, ER quality control (ERQC, endoplasmic reticulum: 10), ER-associated protein degradation (ERAD: 19), chloroplast-associated protein degradation: one, autophagy: 20, proteolysis: 106, protein repair: one) (Appendix A). From a global perspective, we observed that 35 genes out of 80 in the category of multi-process signaling were differentially regulated between Ilpum and NIL (16 genes were downregulated in Ilpum (Log_2_FC: −0.11 to −3.12) but upregulated in NIL (Log_2_FC: 0.03–2.8), and 19 genes were upregulated in Ilpum (Log_2_FC: 0.01–2.71) but downregulated in NIL (Log_2_FC: −0.09 to −2.86), whereas 45 genes recorded a similar transcriptional level (Appendix A).

### 2.8. Differentially Expressed Genes Associated with Transcriptional Regulation

The regulation of gene expression occurs at different levels of the metabolism of the cell. This important event involves many factors, including proteins, which act alone or in complex with other cellular compounds to regulate gene expression under specific conditions. Our RNA-Seq data show that, upon GRH infestation, 599 DEGs (356 upregulated and 243 downregulated) belonging to 53 transcription regulation complexes and transcription factor (TF) families were either induced or suppressed. Of this number, we can mention the basal transcription regulators such as RNA polymerase I (two) and II (59)-dependent transcription complexes, TIFY (13), and Sigma factor (two), as well as the commonly reported TF families such as C2C2 (32), bZIP (basic leucine zipper: 32), MYB (myeloblastosis: 59), Homeobox (33), and AP2/ERF (ethylene-responsive factor: 31) TF superfamilies, and the GRAS (GIBBERELILIN-INSENSITIVE (GAI), REPRESSOR of GA1-3 (RGA), and SCARECROW (SCR): 15), C2H2-ZF (31), C3H-ZF (19), NAC (27), and WRKY (27), basic helix–loop–helix (bHLH: 28) TF families (Figure 4H). Other TF families included NN-like TF (two), Alpin (six), cysteine proteinase 1 precursor (CPP1: two), EIL (six), heat-shock factor (HSF: 12), MAD/AGL (one), SBP (two), TUB (nine), ARR-B (three), and mTERF (14), among others (Figure 4H). Interestingly, the top 10 upregulated (Log_2_FC: 6.24–9.98) and top 10 downregulated (Log_2_FC: −3.07 to −7.78) TF-encoding genes belonged to the Homeobox, GRAS, Alfin, B3, RNA polymerase II, and C2H2-ZF TF families, and the Homeobox, mitochondrial transcription termination factor (mTERF), auxin-responsive factor (ARF), plastid-encoded bacterial-type RNA polymerase (PEP), C2H2F, myeloblastosis (MYB), NAC (no apical meristem (NAM), ATAF12, and cup-shaped cotyledon (CUC2)), polymerase complexes, and nuclear factor Y (NF-Y) TF families, for upregulated and downregulated, respectively.

In addition, among the 599 DEGs associated with the transcriptional regulation in response to GRH-induced biotic stress, 224 genes were differentially regulated between Ilpum and NIL (143 genes were downregulated in Ilpum (Log_2_FC: −0.03 to −7.21) but upregulated in NIL (Log_2_FC: 0.01–9.99), and 81 genes were upregulated in Ilpum (Log_2_FC: 0.01–8.42) but downregulated in NIL (Log_2_FC: −0.1 to −5.99)). However, 375 genes exhibited similar expression patterns between Ilpum and NIL (Figure 6, Appendix A).

### 2.9. Differentially Expressed Genes Associated with Other Complex Cell Metabolism

We observed that GRH-induced biotic stress provoked changes in various plant metabolisms, in addition to what has been described in the previous paragraphs. As shown in panels F–O of Appendix A, hundreds of genes associated with cell-wall organization (130, 95 upregulated: Log_2_FC 0.025–10.05 and 35 downregulated: Log_2_FC −0.04 to −2.92), vesicle trafficking (179), protein translocation (87), solute transport (434), amino-acid metabolism, lipid metabolism (191), nucleotide metabolism (41), coenzyme metabolism (127), chromatin organization (117), and RNA processing (219) were differentially regulated between Ilpum and NIL in response to GRH. To this number, we can add genes involved in cytoskeleton organization (93), protein biosynthesis (233), and DNA damage response (22). Appendix A provides details comparative transcriptome profile based on MapMan characterization between Ilpum and NIL.

### 2.10. Confirmation by qPCR of GRH-Mediated Transcriptional Changes 

With the purpose of validating the RNA-Seq data, the expression of 12 genes selected from RNA-Seq data (Ilpum upregulated genes: Os01g58100 (polyphenol oxidase, POD), Os04g43800 (phenylalamine-lyase), Os05g35290 (phenylalanine ammonia-lyase) and downregulated by GRH: Os03g03360 (ribosomal protein L5), Os06g40640 (fructose-bisphosphate aldolase isozyme), Os09g16910 (cysteine desulfurase 1, mitochondrial precursor); NIL upregulated genes: Os01g62020 (NAD dependent epimerase/dehydratase family domain-containing protein), Os04g43800 (phenylalanine ammonia-lyase), Os12g37260 (lipoxygenase 2.1, chloroplast precursor) and downregulated by GRH: Os01g62244 (ubiquitin-conjugating enzyme), Os07g04840 (23 kDa polypeptide of photosystem II, PsbP), Os12g17600 (ribulose bisphosphate carboxylase small chain, chloroplast precursor) among the most upregulated and downregulated were validated by qPCR (Figure 7A–D).

In the same perspective, we were interested in investigating the expression of a set of genes either showing an enhanced transcript accumulation level in NIL (carrying *Ghr1* locus) compared to Ilpum or exhibiting differential expression patterns between the two genetic materials under study. It was interesting to see that the expression level of Os07g34520 (*OsADF3*, Isocitrate lysate), and that of Os10g37340 (cystathionine gamma-synthase, pyridoxal phosphate-depended enzyme domain-containing protein) significantly increased in NIL (RNA-Seq: 1.93 Log_2_FC, qPCR: 1.86 (24 h) and 1.42 (48 h) for *OsADF3* and RNA-Seq: 1.22 Log_2_FC, qPCR: 0.99 and 0.94 Log_2_FC, at 24 h and 48 h for cystathionine) carrying the resistance QTL *Grh1* compared to Ilpum (RNA-Seq: 0.20, qPCR: 0.23 (24 h) and 0.78 (48 h) Log_2_FC for *OsADF3* and RNA-Seq: 0.11 Log_2_FC, qPCR: 0.13 (24 h) and 0.42 (48 h) Log_2_FC) in response to GRH infestation (Figure 8A,B). In contrast, another set of genes including Os02g43330 (*OsHOX24*, Homeobox-associated leucine zipper), Os03g60580 (*OsWCOR79*, acting-depolymerizing factor), Os01g74450 (Aquaporin), and Os01g06740 (*OsRIP1* or *OsjRIP1.1*, encoding ribosome-inactivating protein II) exhibited differential expression patterns (downregulated in Ilpum but upregulated in NIL) under the same conditions (Figure 8C–F).

## 3. Discussion

### 3.1. Green Rice Leafhopper-Induced Stress Activates Defense-Related Genes and Multiple Regulatory Pathways in Rice

Plants have evolved sophisticated, highly coordinated, and well-structured innate immune system, capable of detecting invading organisms and halting them before they are able to cause extensive damage [38,39]. At a cellular level, upon biotic stress induction, a variety of signaling cascades are activated, which in turn help the plant to localize the source of the stress in order to mobilize the relevant resources and activate the appropriate defense system to combat the stress, while tending to maintain a balanced growth and development [40]. In the process, stress-responsive compounds accumulate in the cell, and a transcriptional reprograming occurs, resulting in the induction or suppression of defense-related genes and the synthesis of proteins. Under the same conditions, secondary metabolites accumulate, and antioxidant systems (enzymatic and nonenzymatic) are activated, as part of the defense mechanism. Here, MapMan analysis revealed that a basal defense system similar to that observed during pathogen attacks was typically activated upon GRH infestation. We can mention, among others, the pathogen-pattern-triggered immunity (PTI: *FLS2* (LOC_Os04g52780, the flagellin receptor protein kinase, upregulated by Log_2_FC 2.7 in Ilpum and 3.45 in NIL) and *PCRK*: LOC_Os10g30600, the bacterial elicitor response protein kinase, which were upregulated in Ilpum and NIL by Log_2_FC 0.9 and 2.3, respectively. In addition, the *NPR1* (nonexpressor of pathogenesis-related gene 1, LOC_Os01g09800), a key regulator of salicylic acid (SA)-mediated resistance and systemic acquired resistance (SAR) [41,42], was upregulated by Log_2_FC 0.87 in Ilpum and 2.61 in NIL. Furthermore, the *TGA* (LOC_Os01g17260, transcription *TGA6* or *AtbZIP45*, a negative regulator of resistance (NRR) proteins [43] similarly downregulated in Ilpum and NIL) was earlier reported to interact with NRP1 while co-expressing with *NRP1* homologs *NH1* and *NH3*, and three *OsWRKY33* TF-dependent plant immunity (similarly upregulated in Ilpum and NIL) and Virus tobamovirus multiplication were the most abundant (Figure 4G). These genes are thought to contribute to the response mechanisms of rice towards GRH resistance.

It was interesting to see that, in addition to biotic stress-related genes, GRH triggered significant changes in the expression of genes previously associated with abiotic stress response in plants. These include the salt overly sensitive (*SOS2*, LOC_Os01g35184) signaling pathway genes, earlier suggested to positively regulate the salt stress response in plants [44], which were found to be upregulated in Ilpum but downregulated in NIL. In the same way, the gibberellin–abscisic acid signaling pathway crosstalk *OsGAI* TF (LOC_Os03g49990, a gibberellin insensitive encoding a DELLA protein) was differentially regulated between Ilpum (downregulated) and NIL (upregulated). This applies also to *COLD1* (LOC_Os04g51180, cold sensor), which was upregulated in Ilpum, while being downregulated in NIL. Another group of abiotic stress-responsive genes included stomata closure (drought), heat response transcriptional regulator and temperature sensing and signaling, and *CAS* (LOC_Os02g49680, extracellular calcium sensor receptor: upregulated in Ilpum but downregulated in NIL) among others. A study conducted by Rolly et al. [45] proposed that phytohormone biosynthesis or signaling pathway genes such as auxin carriers (PIN-FORMED proteins), *MAX1–4* (more axillary branching, strigolactone biosynthesis genes), gibberellin biosynthesis genes (*GA20ox1* and *2*), and cytokinin biosynthesis (isopentenyl transferase, *IPT5* and *7*) could be involved in the adaptive response towards drought tolerance in *Arabidopsis*.

It is well established that photosynthesis and respiration are two fundamental physiological processes required for the survival of plants during stress [46]. In this study, Appendix A, BIN 1 summarizes the differential transcriptome profile between Ilpum and NIL. A recent report showed that ATP, NADPH, and carbohydrates are generated as part of photosynthesis, and these resources are then utilized for the synthesis of many other compounds such as primary metabolites and defense-associated hormones (abscisic acid, ABA; ethylene, ET; jasmonic acid, JA; salicylic acid, SA) [47]. Other studies have supported that both an increase and a decrease in photosynthesis rates can occur, and photosynthesis can be assumed as a source of energy for plant defense against stress inducers [48]. Considering the expression patterns of genes associated with the photosynthesis, during GRH feeding on rice seedlings, where the NIL (carrying *Grh1*) recorded the highest number of downregulated genes (16.5%, exceeding that observed in Ilpum), it appears that Ilpum experienced a high level of stress caused by GRH compared to NIL. The NIL (GRH resistant) might have activated other defense systems providing the required cellular response towards GRH resistance. We could then propose that, during insect pest feeding on host plants, the photosynthesis rate increased to provide enough energy as part of the primary metabolism, which contributed to the adaptive response mechanisms towards insect pest resistance and helped maintain balanced growth and development of plants. 

In the biological system, biological processes such as photosynthesis and cellular respiration have evolved multiple biochemical steps, which occur simultaneously within the cells and share common molecular components [49]. Here, we observed that the majority of genes (61.6%) involved in carbohydrate metabolism and expressed in response to GRH were downregulated in Ilpum vs. 53.8% upregulated in NIL (Appendix A, BIN 3). Cellular respiration is central to the life of plants. In the same way, 67.4% and 52.6% of genes involved in cellular respiration were upregulated in Ilpum and NIL, respectively, vs. 32.6% and 47.4% showing a downregulation pattern. Likewise, secondary metabolites such as terpenoids and flavonoids, among others, have been shown to contribute to boosting the plant immune system in response to abiotic or biotic stress [50]. Looking at the RNA-Seq data and MapMan analysis results, Ilpum and NIL showed similar expression patterns of genes involved in secondary metabolism (26.6% and 73.4% down- and upregulated in both backgrounds). This is also true for genes involved in DNA damage response (Appendix A, BIN 14). Therefore, we could then argue that rice plants exposed to GRH infestation would experience a disturbed photosynthetic process, while witnessing an enhanced cellular respiration and accumulation of secondary metabolites as part of the defense mechanisms towards GRH resistance.

Maintaining genome integrity is recognized as crucial for all living organisms, especially under stress conditions where plants experience changes in their metabolism and increased accumulation of bioactive molecules such as reactive oxygen species (ROS) that may cause DNA damage and cell death [51]. Our data showed that RNA biosynthesis genes were 51% downregulated vs. 59.4% upregulated in Ilpum and NIL, respectively. Among RNA processing genes, Ilpum and NIL had similar expression patterns of transcription factor-encoding genes (61.6% and 64.4% downregulated in Ilpum and NIL, respectively; 38.4% and 35.6% upregulated in Ilpum and NIL, respectively) (Appendix A, BIN 15 and BIN 16).

Furthermore, the transcription profile of genes related to amino-acid metabolism (BIN 4), lipid metabolism (BIN 5), coenzyme metabolism (BIN 7), polyamine metabolism (BIN 8), protein modification (BIN 18), and protein homeostasis (BIN 19) indicated that the number of downregulated genes by GRH in these categories was much higher in Ilpum compared to the GRH-resistant rice line (NIL), whereby some of the downregulated genes in Ilpum were differentially regulated under the same conditions. 

Generally, genes whose expression is highly induced upon stress induction (abiotic or biotic stress) are considered as having a high potential to be involved in the adaptive response or resistance mechanisms towards that particular stress. In addition, most upregulated and downregulated genes are thought to have differential contributions acting as positive or negative regulators of plant defense. In this study, among the top 20 DEGs between Ilpum (GRH susceptible) and NIL (carrying *Grh1*), pooled from the GRH-mediated transcriptome data (Table 1), six were previously characterized, including *OsMRG702* (LOC_Os11g34300, a reader protein of H3K4me3 [52], upregulated in Ilpum (9.6 Log_2_FC) but downregulated in NIL (−0.44 Log_2_FC)), *OsAIP1* or *OsBAT1* (LOC_Os01g36890, involved in biotic stress response [53], upregulated in Ilpum (9.6 Log_2_FC) but downregulated in NIL (−4.6 Log_2_FC)), and *OsFtsH2* (LOC_Os06g45820) identified as cell division protease, homolog of *AtFtsH2*/8 [54], upregulated in Ilpum (8.2 Log_2_FC) but downregulated in NIL (−0.92 Log_2_FC). In contrast, genes such as *OsBIHD1* (LOC_Os03g03g47740, homeodomain containing protein) [55,56,57], *OsRLI* (LOC_Os11g34350, RNA L inhibitor belonging to ATP-binding cassette (ABC) sub-family E member 1 [58]), and *OsFNR* (LOC_Os03g57120, ferredoxin-NADP reductase chloroplast precursor) [59]), previously reported as being involved in biotic stress resistance in rice, were downregulated in Ilpum (Log_2_FC −0.63 (*OsBIHD1*), −024 (*OsRLI*), and −0.13 (*OsFNR*)), while being highly induced in NIL (Log_2_FC 9.99 (*OsBIHD1*), 8.79 (*OsRLI*), and 8.39 (*OsFNR*)) under the same conditions. 

### 3.2. Complex Signaling Networks Activated upon GRH Infestation in Rice

The molecular mechanisms against the attack of insect pests, including GRH, involve a wide range of complex signaling networks, predominantly mediated by but not limited to jasmonic acid (JA) [60,61]. Other phytohormones such as ethylene (ET) and salicylic acid (SA) have been shown to be activated in the process [40,62,63,64,65,66] and trigger changes in the transcript levels of stress-responsive genes. Under the same conditions, ROS and RNS accumulate, which have a dual effect; at a low level, they serve as signaling molecules and help plants to quickly react and activate the required defense system, whereas, when over-accumulated, ROS or RNS and derivative compounds are detrimental to plant fitness [67,68]. The reduction–oxidation (redox) homeostasis of the plant is compromised when these ROS and RNS over-accumulate or are in imbalance, thus causing oxidative or nitro-oxidative stress, lipid peroxidation denoting cell membrane degradation, and eventually culminating to cell death. Thus, considering that a high number of genes associated with redox homeostasis were downregulated by GRH in NIL, in contrast with those upregulated in Ilpum (Appendix A, BIN 10), we could say that a balanced redox was maintained in the GRH resistant line NIL, which experienced a low level of stress as compared to the susceptible rice cultivar Ilpum, eventually due to over-accumulation of ROS or RNS that have the ability to cause oxidative damage and cell death.

Respiratory burst oxidase homologs (RBOHs) are recognized as being crucial for defense signaling events and are key players in plant immunity [69,70,71,72,73], allowing the activation of antioxidant systems (nonenzymatic and enzymatic antioxidants), including catalase (CAT) [74,75], glutaredoxin (Grx) [76,77,78], and superoxide dismutase (SOD) [79,80], to combat the stress and give a robust defense response. In this perspective, we could speculate that the significant upregulation of RBOH encoding genes (LOC_Os05g45210, LOC_Os09g26660, LOC_Os09g26660, LOC_Os01g53294, LOC_Os01g25820) in Ilpum (1.8–5.03 Log_2_FC), while being either downregulated or not affected in NIL (0.08 to –2.05 Log_2_FC), coupled with the differential transcript accumulation of ROS scavenger-related genes (LOC_Os02g02400 (catalase, *CAT-A*), LOC_Os06g51150 (*CAT-B*), LOC_Os03g03910 (CAT isozyme 2), LOC_Os07g46990 (superoxide dismutase, *SOD*)) and hydrogen peroxide removal-related genes (ascorbate peroxidase (LOC_Os04g35520) and ascorbate free radical reductase (LOC_Os02g47800)), among others, gives insights into the observed enhanced resistance of NIL plants to GRH and a low level of ROS accumulation compared to Ilpum, while suggesting a possible interaction with *Ghr1.*

### 3.3. Differential Feeding Behaviors of GRH between Ilpum (Susceptible) and NIL (Carrying Grh1) Give Insights into Their Survival and Transcriptome Profiles in Rice

Studies on GRH resistance in plants have proposed that rice varieties showing resistance to GRH possess three characteristics of insect resistance, namely, antibiosis [81,82], antixenosis [83], and tolerance [84], causing insect high mortality, slow development, and infecundity, among others. In addition, a study conducted by ABE [85] investigated the mechanism of varietal resistance to GRH (*Nephotettix cincticeps* Uhler) found that, when leafhoppers were reared on resistant plants, their development was slowed down, followed by a rapid death, especially first nymphs. The author also indicated that the sugar content of the excreta was roughly below 0.1% in GRHs reared on resistant rice varieties, contrasting with nearly 0.8–1.6% of sugar content of the excreta of GRHs reared on susceptible rice varieties. In a converse approach, Reddy et al. [86] supported that the feeding by planthoppers on susceptible rice varieties caused a reduction in stored sugar content. In the same way, Heong and Hardy [87] reported that sugar content, such as sucrose (predominant) [82], glucose, fructose, or maltose, is required for successful development and fitness of planthoppers (in this case brown planthopper, BPH). We could then say, on the one hand, that the low survival rate (8%) of second nymph GRHs infested in NIL (carrying *Grh1*) compared to the 72% survival rate recorded in Ilpum 72 h after infestation would imply that GRHs reared on the resistant NIL seedlings starved to death due to a failure to suck the phloem sap of NIL (Nc4, Figure 1B), whereas, in the susceptible rice variety (Ilpum), GRHs were able to suck the phloem sap (Nc6, Figure 1A) containing the required sugar for their fitness and development. However, we suspect that GRHs may not have penetrated the phloem and taken up nutrients from host plants considering the activation of defense mechanisms in NIL carrying *Grh1*, coupled with the evident boost in the transcriptional regulation of defense-related genes. On the other hand, it is thought that the failure to penetrate the stylet and feed from the phloem of NIL could be attributed to the robust defense system activated, coupled with the active signaling pathway-related genes and antioxidant-associated protein-coding genes, and their interplay contributed to the observed resistance of NIL compared to Ilpum.

Therefore, considering the transcriptome profile of GRH-responsive genes in Ilpum and NIL, and the differential expression patterns displayed in the two rice genetic materials, coupled with the interesting functions of genes with a log_2_ fold change value above 2, data generated by this study suggest that *Grh1*-mediated resistance against green rice leafhopper (GRH) is a result of a complex and a combinational action of transcriptome-wide interactions, coupled with biochemical reactions and physiological processes across the plant, rather than gene–gene action. In addition, RNA-Seq and ManMan data associated with the EPG results propose that the feeding behavior of GRHs on rice plants determines the kind of defense mechanism and physiological processes that will be activated, as well as the biochemical reactions that will take place, in order to provide a proper response towards the stress, while tending to maintain balanced plant growth and development.

## 4. Materials and Methods

### 4.1. Green Rice Leafhopper Growth Conditions

Prior to infestation, second nymph GRHs were grown and maintained as described previously [33]. Briefly, GRHs were reared at 25 °C, under a 16 h light and 8 h dark cycle and 60–70% relative humidity, in cages of 50 cm × 50 cm × 50 cm, width, length, and height, respectively, with each cage containing GRHs of similar growth stage (nymphs to adults). The development cycle of GRH takes about 25 days from eggs to adult insects, distributed as follows: first to fourth instar nymphs accounting for 25 days (first to second instar nymphs: 10 days, second to third instar nymphs: 5 days, third to fourth instar nymphs: 5 days, and fourth to adult: 5 days). At each development stage, GRHs were reared in different cages, and second instar nymphs GRH were used for downstream experiments.

### 4.2. Plant Growth Conditions and Green Rice Leafhopper Infestation

Two rice lines, Ilpum (the GRH-susceptible cultivar) and a near-isogenic line (NIL) carrying the *Grh1* (a gene conferring resistance towards GRH) derived from a cross between Shingwang (*indica*, P1) and Ilpum (*japonica*, P2) [33], were use as genetic materials for RNA-Seq analysis. In addition, the *Grh1* donor parent Shingwang was included in the study solely to monitor the feeding behavior of GRH during the electrical penetration graph (EPG) test. Prior to germination, rice seeds were surface-sterilized using nitric acid (HNO_3_, 0.7%) (CAS: 7697-37-2, Junsei Chemical Co. Ltd., Tokyo, Japan) overnight to break the dormancy [88], followed by incubation for 48 h at 27 °C to induce germination. Then, germinated rice seeds were grown in 50-well trays (one germinated seed per well) containing an enriched soil supplemented with basic nutrients necessary for rice seedling growth as indicated by the manufacturer (Boonong, Gyeongju, Korea, humidity: 18–30%, pH: 4.5–5.5, EC: below 2.0) until three-leaf stage. Seedlings with a uniform height (one seedling per rice line per test tube, in quintuplets for control and GRH-infested seedlings) were placed in test tubes after washing the loose soil from the roots. Seedlings were infested with second instar nymphs green rice leafhoppers (10 GRHs per test tube per seedling in quintuplet) for 48 h at room temperature as described previously [33]. Leaf samples for RNA-Seq analysis were collected in duplicate (from two different seedlings) from control (GRH non-infested seedlings) and GRH-infested seedlings, before being snap-frozen in liquid nitrogen and sent for sequencing (Macrogen, Daejeon, Korea). 

### 4.3. Electrical Penetration Graph (EPG) Test and GRH Survival Rate

We monitored the feeding behavior of GRH on selected plant materials using an electrical penetration graph (EPG) technology as described previously [34]. In essence, a stylet probing the behavior of waveforms in each process from host detection to insect feeding was used to examine the interaction between phloem-feeding activities of the insect on the host plant. In the process, different waveforms are recorded and are classified from Nc1 to Nc6. Nc1–4 waveforms indicate that the insect (GRH) sucked only the xylem sap, while the Nc6 waveform indicates that the green rice leafhopper sucked the phloem sap [35,89]. 

The EPG test was conducted on male GRHs initially anesthetized with carbon dioxide (CO_2_), and a gold line electroductive silver conductive paint (P-100, CANS, Japan) was attached to GRH wings. The gold lines were connected to the GIGA-4 DC EPG amplifier (Washington University, The Netherlands). EPG waveforms were analyzed using PROBE 3.4 software (W.F. Tajallingii, Wageningen University, The Netherlands). The substrate voltage was adjusted in the range of +5 V to −5 V [89,90]. Male GRHs were inoculated to the third-leaf stage of rice, and the waveforms were recorded for 6 h. All insects were recorded simultaneously in 10 replicates for the EPG test, with Nc indicating the waveforms of GRH (*Nephotettix cincticeps* Uhler). The mean waveform duration (Nc1–Nc6), mean duration per waveform event, and mean number of waveform events per insect were calculated (Table 3).

The survival rate of GRH in Ilpum and NIL GRH-infested seedlings was determined over time as follows: (total number insects (GRH) initially infested—number of dead insects over time)/(total number of alive insects initially infested) × 100, and the data were expressed as a percentage.

### 4.4. RNA-Seq, Data Analysis, and Quality Control

Total RNA was isolated from leaf samples using a Takara MiniBEST RNA extraction kit (CAS No 9767, Japan), and DNA contamination was eliminated using DNase. An mRNA purification kit was used to prepare the library according to the manufacturer’s protocol (Library kit: TruSeq Stranded Total RNA LT Sample Prep Kit (Plant); protocol: TruSeq Stranded Total RNA Sample Prep Guide, Part #15031048 Rev.E; Reagent: TruSeq 3000 4000 SBS Kit v3). RNA fragments were randomly purified for short-read sequencing, followed by reverse transcription of fragmented RNA into cDNA. The adapters were ligated onto both ends of the cDNA fragments. After amplifying fragments using PCR, fragments with insert sizes between 200 and 400 bp were selected, and both ends of the cDNA were sequenced by read length (paired-end sequencing) (Macrogen, Beolkkot-ro, Geumcheon, Seoul, Korea) using Illumina’s HiSeq™ 4000 Sequencing System facilitated by HCS 3.3.52 software (Illumina, Inc. 9885 Towne Centre Drive, San Diego, CA 92121, USA) according to the manufactures’ instructions (Sequencing protocol: HiSeq 3000 4000 System User Guide Document#15066496 v05 HCS 3.3.52). 

The quality control of the sequenced raw reads was done via a computational method using the FastQ Screen program (version 0.3, Babraham Institute, Cambridge, UK) [91] to ensure overall read quality, total bases, total reads, and GC (%). In order to reduce biases in analysis, artefacts such as low-quality reads, adaptor sequences, contaminant DNA, or PCR duplicates were removed, and the trimmed reads (Trimmed Mean of M-values, TMM, normalized with *edgeR* package in R (version 4.1.1, USA) for all samples were compared with each other and merged into one file prior to performing the transcriptome assembly. Merged data were assembled using Trinity (http://trinityrnaseq.github.io), accessed on 25 August 2021, to form contigs (longer fragments without N gaps). 

### 4.5. Clustering Transcripts into Unigenes and ORF Prediction

For assembled genes, the longest of the assembled contigs were filtered and clustered into nonredundant transcripts using CD-HIT-EST [92,93]. We defined these transcripts as “unigenes”, which were used for predicting the ORFs (open reading frames), annotating against several known sequence databases and analyzing differentially expressed genes (DEGs). ORF prediction for unigenes was performed using the TransDecoder program to identify candidate coding regions within the transcript sequence. After extracting ORFs that were at least 100 amino acids long, TransDecoder predicted the likely coding regions.

### 4.6. Raw and Trimming Data Statistics

The total number of bases, reads, GC (%), Q20 (%), and Q30 (%) were calculated for four samples. The Trimmomatic program [94] was used to remove adapter sequences and bases with base quality lower than 3 from the ends. Using the sliding window method, bases of reads that did not qualify for a window size of 4 and a mean quality of 15 were trimmed. Afterward, reads with length shorter than 36 bp were dropped to produce trimmed data.

### 4.7. Annotation on Gene Ontology and EggNOG Databases

For functional annotation of the unigenes, the Gene Ontology (GO) database was applied to classify the annotated unigenes using BLASTX of DIAMOND with an E-value cutoff of 1.0 × 10^−5^. Classification of GO terms was subsequently performed using and in-house script. The GO terms belonging to biological processes (BPs), cellular components (CCs), and molecular function (MFs) were listed (Appendix A). 

To identify the proteins distributed in eukaryotic orthologous groups (KOG), clusters of orthologous groups (COGs), and non-supervised orthologous groups (NOGs), we carried out BLASTX of DIAMOND with an E-value cutoff of 1.0 × 10^−5^ in the EggNOG (Evolutionary Genealogy of Genes) database. The annotated unigenes were mapped to the annotation of the corresponding orthologous groups in the EggNOG database.

### 4.8. Annotation on KEGG Pathway, UniProt, and Pfam Databases

For functional annotation of the unigenes, we carried out BLASTX of DIAMOND with an E-value cutoff of 1.0 × 10^−5^ in the Kyoto Encyclopedia of Genes and Genomes (KEGG) database. Bidirectional best hit (BBH), which is a widely used method to infer orthology, was used to search against the KEGG database to obtain the KO (reference pathway) number of the KEGG annotation. The KO number of the transcriptome was also obtained according to KEGG annotation (*Oryza sativa* Japonica RAPDB (RAPDB_IRGSP-1.0) genome database, https://www.genome.jp/kegg-bin/show_organism?org=dosa), accessed on 25 August 2021.

For functional annotation of the unigenes, we carried out BLASTX of DIAMOND with an *E*-value cut-off of 1.0 × 10^−5^ in the UniProt (Universal Protein Resource) database. In addition to capturing the core data mandatory for each UniProtKB entry (mainly the amino-acid sequence, protein name or description, taxonomic data, and citation information), as much annotation information as possible was added.

For functional annotation of the unigenes, we carried out BLASTX of DIAMOND with an E-value cut-off of 1.0 × 10^−5^ in the Pfam database. The Pfam database is a large collection of protein families, each represented by multiple sequence alignments and hidden Markov models (HMMs). Proteins are generally composed of one or more functional regions, commonly termed domains. The identification of domains that occur within proteins can, therefore, provide insights into their function.

### 4.9. MapMan Analysis

Big data with several genes and transcripts often challenge efficient data analysis. Common Gene Ontology (GO) enrichment analysis relying on a database search, although useful for functional characterization of specific gene sets, happens to be less informative if detailed analysis is required at the level of studying genes involved in particular pathways or physiological functions. In this regard, various tools have been developed and used with variable success, e.g., the TM4 suit by Saeed et al. [95], and GoMiner^TM^ by Zeeberg et al. [96].

In addition, the MapMan omics data analysis program (http://mapman.gabipd.org), accessed on 25 August 2021, has been widely used for visualization of omics data at the process or pathway level. The software is designed and optimized to map transcriptomic data in existing databases for different plant species, including rice. MapMan employs a hierarchical BIN-based ontology system (BINs are functional categories), where specific bins are allocated to biological processes or molecular functions and sub-bins are allocated to individual steps or nodes in that particular biological process. For instance, BIN number 26 is related to stress, and BIN numbers 26.1–6 refer to abiotic stress, while BIN numbers 26.8–10 indicate biotic stress. In the same way, sub-bins associated with abiotic stress include 26.1 (light stress), 26.2 (carbon dioxide sensing and signaling), 26.3 (gravity sensing and signaling), 26.4 (26.4.1: temperature sensing; 26.4.2: heat stress; 26.4.3: cold stress), 26.5 (drought stress), and 26.6 (salt stress). Similarly, sub-bins 26.8.1, 26.8.3, 26.9.1, and 26.10.1.2 are associated with pathogen pattern-triggered immunity (PTI), defense mechanisms, symbiont-symbiosis signaling, virus movement, and tobamovirus multiplication, respectively. This approach helps to minimize the redundancy usually found in GO enrichment analysis. Furthermore, MapMan uses expression values of genes and displays the analyzed data in the form of diagrams, which allows a better understanding of the significance of the data, where genes with increased or decreased expression levels are indicated as color-coded square blocks. In the perspective of obtaining meaningful information, 8859 DEGs were analyzed using MapMan 3.6.0RC1. To achieve that, all gene expression data with significant DEGs (*p* < 0.05) were formatted in Microsoft Excel using their unique locus identifiers and expression values expressed as Log_2_FC and saved as a tab-delimited file. The file was further mapped against the rice genome database (X4.2 oryza_sativa) in MapMan. Then, the analyzed data were classified into different metabolic pathways.

### 4.10. Validation of RNA-Seq Data by qPCR

The TaKaRa MiniBEST Universal Plant RNA Extraction Kit (TAKARA Bio Inc., Cat. No. 9769 v201309Da, Kusatsu, Japan) was used to extract the total RNA from leaf samples, following the manufacturer’s instructions. Briefly, frozen leaf samples with liquid nitrogen were crushed to fine powder, and 450 μL of buffer RL (containing 50× dithiothreitol (DTT, 20 μL per 1 mL of buffer RL)) was added and pipetted up and down for a few seconds until the lysate showed no precipitate. The mixture was then centrifuged at 12,000 rpm for 5 min (top bench centrifuge), and the supernatant was transferred to fresh 1.5 mL Eppendorf tubes (e-tubes) (Step 1). Then, one volume of ethanol (100%) was added to the mixture from Step 1, followed by mixing by pipetting up and down, and 600 μL was transferred to a pin column with a 2 mL collection tube. The tubes were centrifuged for 1 min at 12,000 rpm, and the flowthrough was discarded, followed by the addition of 500 μL of buffer RWA and centrifugation for 30 s at 12,000 rpm. Immediately after discarding the flowthrough, 600 μL of buffer RWB was added to the spin column and centrifuged for 30 s at 12,000 rpm (this step was repeated twice). Empty spin columns with collection tubes were centrifuged for 2 min at 12,000 rpm, and the spin column were placed on fresh 1.5 e-tubes; then, 100 μL of RNase free water was added. Finally, samples were incubated for 5 min at room temperature, followed by centrifugation for 2 min at 12,000 rpm (the elution step was repeated twice).

To synthesize the cDNA (complementary DNA) [97], 1 μg of RNA was used as a template, and the ProtoScript^®^II First Strand cDNA Synthesis Kit (New England BioLabs Inc., NEB Labs, MA, USA) was employed according to the manufacturer’s instructions. The newly synthesized cDNA was then used as a template for qPCR (quantitative real-time polymerase chain reaction) to validate the transcript accumulation of the six upregulated (three in Ilpum and three in NIL) and six downregulated (three in Ilpum and three in NIL) genes.

To validate the expression patterns of differentially expressed genes (DEGs) selected from the RNA-Seq dataset, a reaction mixture was prepared comprising 10 μL of Prime Q-Master Mix (with SYBR green I), 0.1× ROX (0.1 μL/50×) (GENETBIO Inc., Daejeon, Korea), 1 μL of template DNA, and 10 pM of each forward and reverse primers in a total reaction volume of 20 μL, including a no-template control (NTC). A three-step reaction including polymerase activation at 95 °C for 10 min, denaturation at 95 °C for 20 s, annealing at 60 °C for 30 s, and extension at 72 °C for 30 s was performed in a real-time PCR machine (QuantStudio^TM^ Design and Analysis Software v.1.3, Applied Biosystems, Thermo Fisher Scientific, Seoul, Korea), with a total of 40 reaction cycles, and the data were normalized to the relative expression of rice Actin1. The list of primers used to validate RNA-Seq results by qPCR can be found in Appendix A.

## 5. Conclusions

Green rice leafhopper (GRH) is a permanent threat to rice cultivation and food security, especially in East Asia. The use of GRH resistance rice lines is the most effective way to overcome crop failure caused by GRH and viral diseases conveyed by this insect pest. This study performed a comparative transcriptome-wide analysis of GRH-responsive genes in GRH-susceptible and -resistant rice genotypes. The RNA-Seq results revealed that phytohormone signaling pathways, carbohydrate metabolism, protein modification and homeostasis, nutrient uptake, and solute transport-related genes, among others, were predominantly upregulated in the GRH-resistant near-isogenic line (NIL, carrying *Grh1* locus). In addition, genes associated with basal defense and systemic acquired resistance (SAR), as well as major transcription factors encoding genes belonging to GRAS, NAC, bHLH, MYB, bZIP, WRKY, and C2H2-F families, were differentially regulated between NIL and Ilpum. Furthermore, the electrical penetration graph revealed that GRHs reared on NIL failed to suck phloem sap and starved to death. However, genes involved in cellular respiration, as well as genes related to photosynthesis, redox homeostasis, secondary metabolism, and external stimulus response, were abundantly induced in the GRH-susceptible rice cultivar Ilpum. Therefore, all data suggest that *Grh1*-mediated activation of multiple regulatory and signaling pathways in response to GRH may involve an interactome with a large number of defense systems towards GRH resistance in rice, rather than gene–gene actions. Advanced functional analyses would help to elucidate the mechanism underlying GRH resistance in rice on a genome-wide scale.

## Figures and Tables

**Figure 1 ijms-22-10696-f001:**
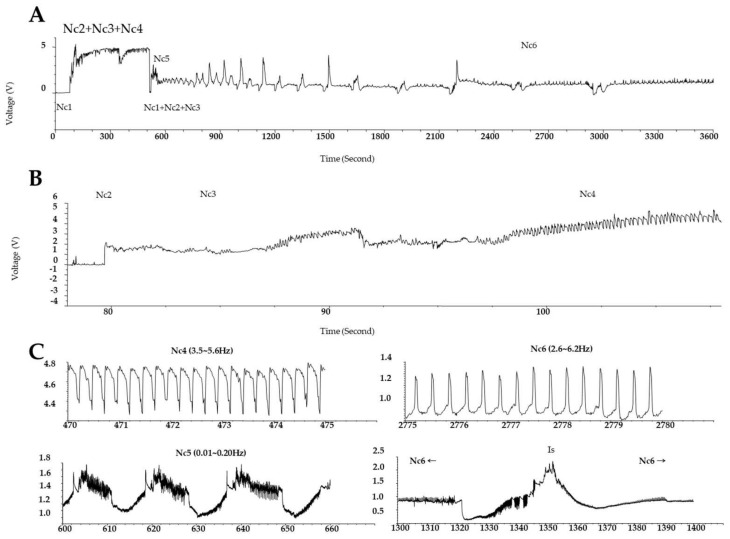
Electrical penetration graph (EGP) showing the feeding patterns of green rice leafhopper over time. EPG of GRH reared (**A**) on Ilpum (GRH-susceptible) and (**B**) on near-isogenic line (NIL) harboring *Grh1* locus; (**C**) frequencies of waveforms (Nc1 to Nc6). Typically, to take up phloem sap, waveforms are categorized as follows: Nc1→Nc2→Nc3→Nc5→Nc6. Nc1 means non-probing, Nc2 indicates the penetration of stylet into host, Nc3 signifies path and salivation, Nc4 represents xylem feeding, Nc5 shows salivation or physical activity in the phloem, and Nc6 represents phloem sap sucking. Nc2–Nc6 indicate the time to reach the stylet to phloem and ready to take up nutrients from the phloem. Accumulated Nc6 means the accumulative time to take up nutrients from the phloem. Shingwang and NIL showed similar patterns in Nc2–Nc6, accumulated Nc6, and Nc2–Nc5.

**Figure 2 ijms-22-10696-f002:**
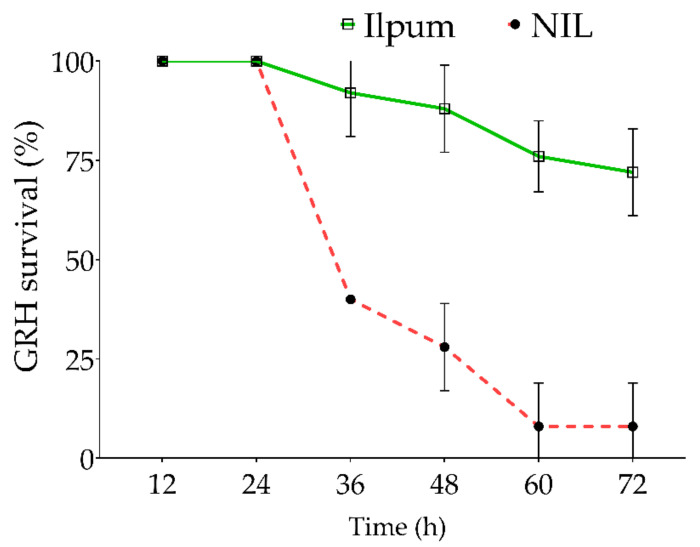
Survival percentage of green rice leafhopper over time. Ten green rice leafhoppers (GRHs) were placed in each test tube containing a single rice seedling. The number of alive GRHs was counted every 12 h from the initial infestation time, and the survival percentage was estimated for each time point. The red dotted line indicates the near-isogenic line (NIL, a cross between Shingwang carrying *Grh1* locus and Ilpum) GRH-infested, while the green continuous line represents the survival of Ilpum GRH-infested. Error bars denote the SD of mean values (10 insects per tube) in quintuplet.

**Figure 3 ijms-22-10696-f003:**
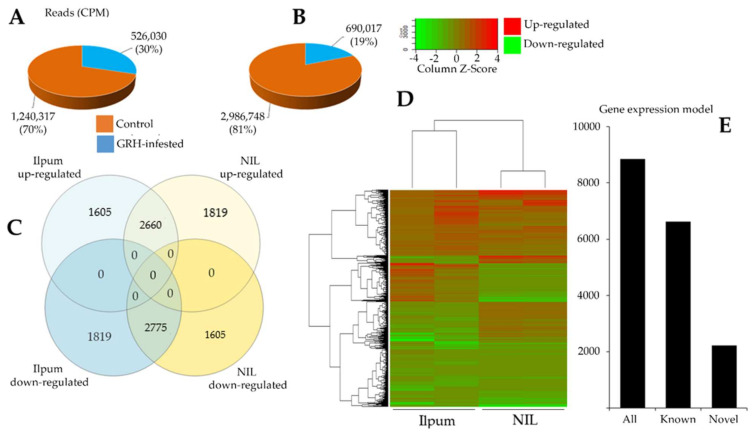
Differentially expressed genes of green rice leafhopper (GRH)—responsive genes between Ilpum and near-isogenic line (NIL). Reads count in (**A**) Ilpum and (**B**) NIL control and GRH-treated plants; (**C**) up- and downregulated genes in Ilpum and NIL; (**D**) heat map showing differentially expressed genes (DEGs) between Ilpum and near-isogenic line (NIL) in response to GRH in duplicate (two samples sequenced per rice line for nontreated and GRH-infested seedlings. Data used to generate the heat map are expressed as Log_2_FC); (**E**) gene expression model. Log_2_FC, logarithm base 2 of the fold change calculated from the fragment per kilobase of transcript per million mapped reads (FPKM Value_2/Value_1).

**Figure 4 ijms-22-10696-f004:**
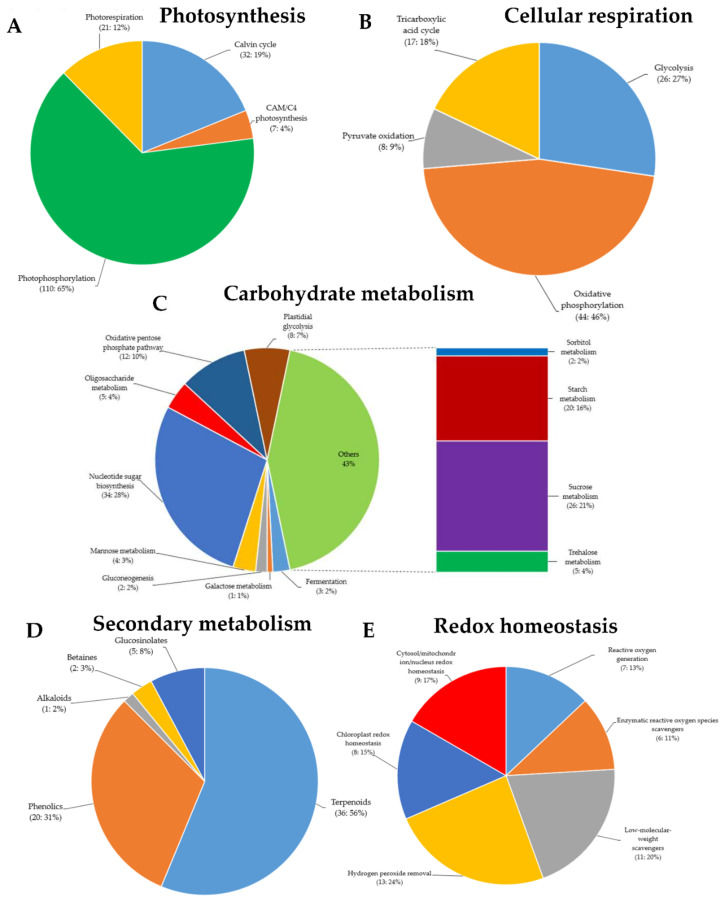
Gene Ontology (GO) terms of green rice leafhoppers fed rice. Differentially expressed genes (DEGs) associated with (**A**) photosynthesis, (**B**) cellular respiration, (**C**) carbohydrate metabolism, (**D**) secondary metabolism, (**E**) reduction–oxidation (redox) homeostasis, (**F**) phytohormone biosynthesis and signaling, (**G**) external stimuli response, and (**H**) transcriptional regulation in response to green rice leafhopper (GRH)-induced biotic stress in rice.

**Figure 5 ijms-22-10696-f005:**
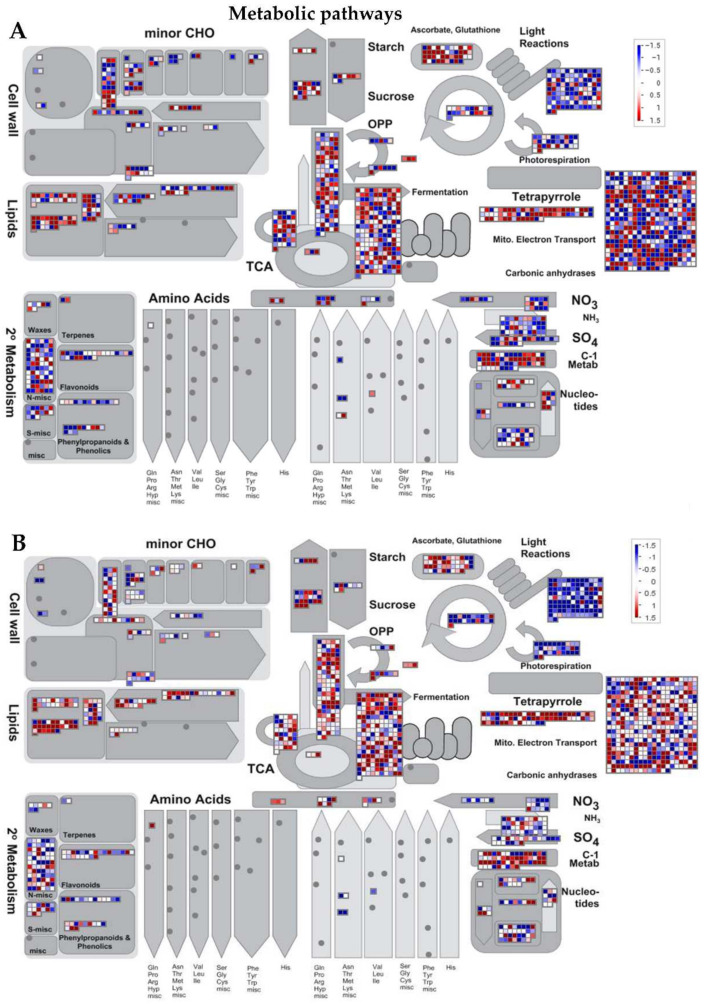
MapMan illustration of differentially changes in transcriptional regulation of metabolic pathways DEGs between Ilpum and NIL in response to GRH infestation. Induction or suppression of the expression of genes involved in different metabolic pathways in (**A**) Ilpum and (**B**) near-isogenic line (NIL) derived from a cross between Ilpum (recurrent parent) and Shingwang (donor parent of *Grh1*), in response to green rice leafhopper (GRH) infestation in rice. Red squares are upregulated genes and blue squares are downregulated genes. Data are logarithm base 2 of fold change (Log_2_FC) (FPKM_value 2/FPKM_value 1). Red boxes are upregulated genes, blue boxes are downregulated genes, and white color indicates Log_2_FC ± zero.

**Figure 6 ijms-22-10696-f006:**
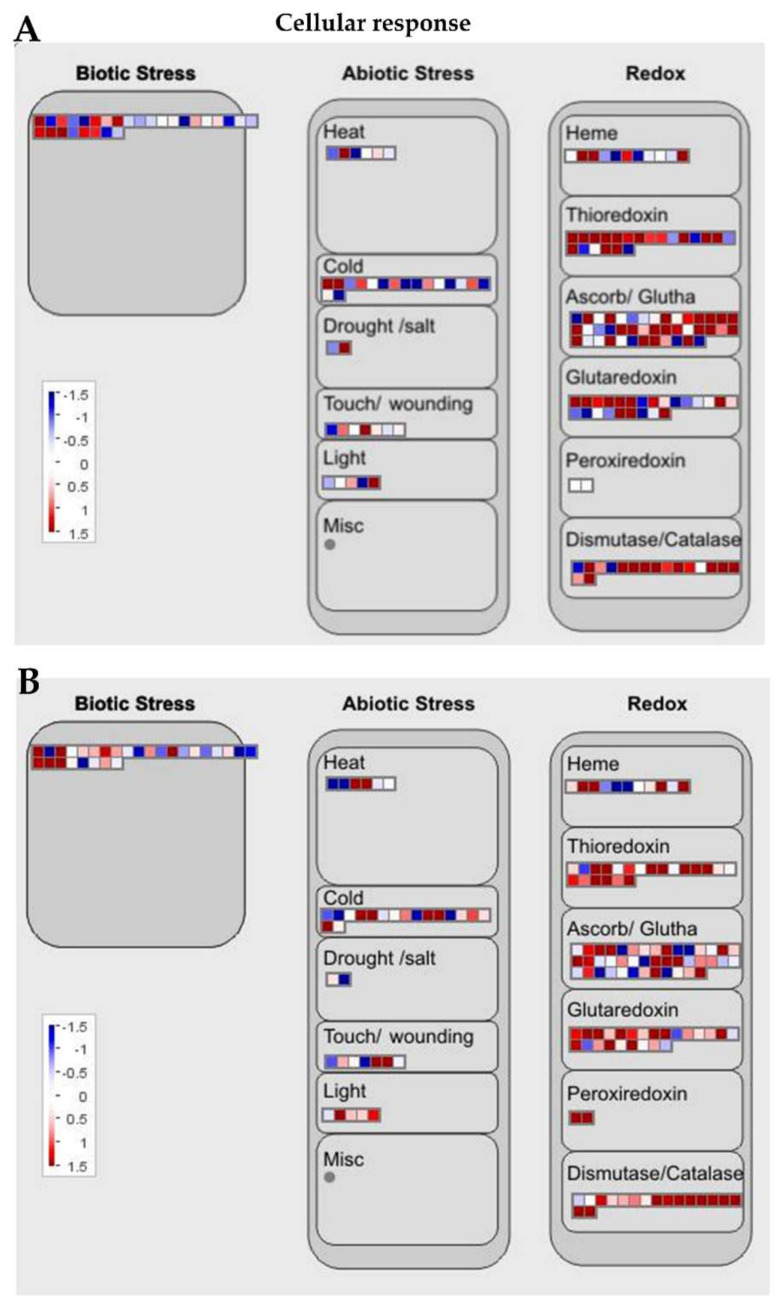
MapMan cellular response overview of the transcriptional regulation between Ilpum and NIL upon GRH infestation. Induction or suppression of the expression of genes involved in different metabolic pathways in (**A**) Ilpum and (**B**) near-isogenic line (NIL) derived from a cross between Ilpum (recurrent parent) and Shingwang (donor parent of *Grh1*), in response to green rice leafhopper (GRH) infestation in rice. Red squares are upregulated genes and blue squares are downregulated genes. Data are logarithm base 2 of fold change (Log_2_FC) (FPKM_value 2/FPKM_value 1). Red boxes are upregulated genes, blue boxes are downregulated genes, and white color indicates Log_2_FC ± zero.

**Figure 7 ijms-22-10696-f007:**
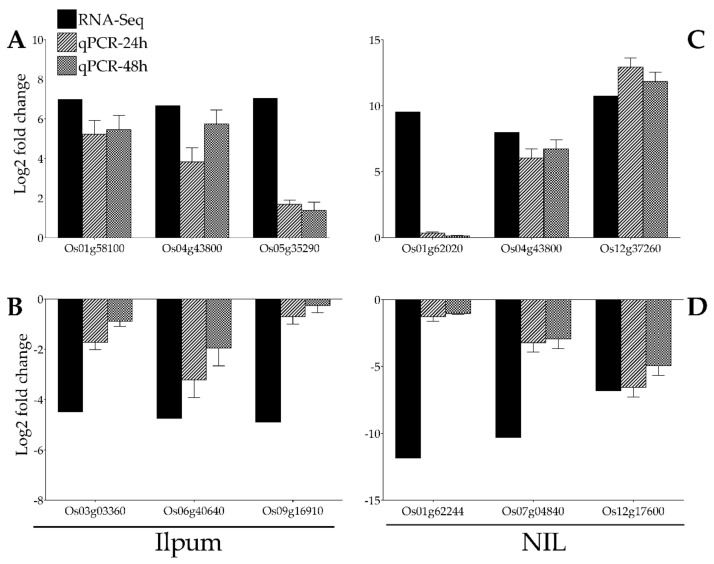
Confirmation of RNA-Seq results by qPCR. A total of 12 genes identified as GRH-responsive through RNA-Seq-based transcriptome analysis (black bars) in GRH-infested rice leaves from Ilpum and derived near-isogenic line (NIL) were selected to validate the fold change in their expression by qPCR (bars with motif filling). These genes belong to different functional groups involved in various metabolic pathways in rice. (**A**) Upregulated and (**B**) downregulated genes in Ilpum, and (**C**) upregulated and (**D**) downregulated genes in NIL in response to GRH infestation.

**Figure 8 ijms-22-10696-f008:**
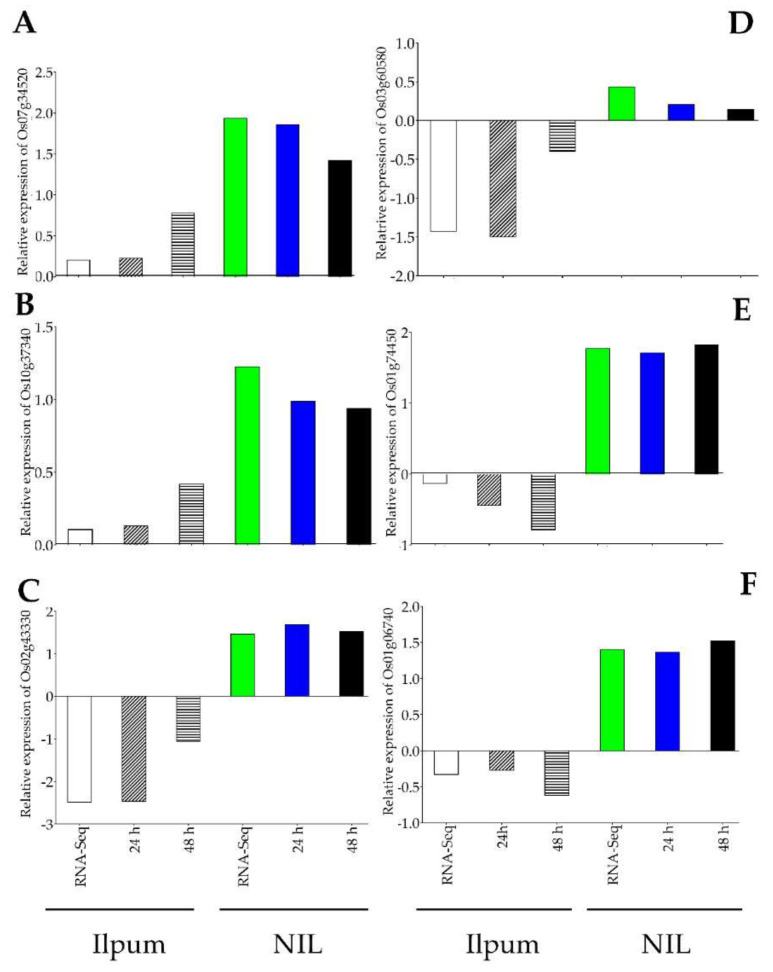
qPCR validation of differentially expressed genes between Ilpum and NIL. (**A**) Enhanced transcript accumulation of Os07g34520 and (**B**) Os10g37340 in NIL compared to Ilpum in response to GRH feeding. (**C**) Differential expression of Os02g43330, (**D**) Os03g60580, (**E**), Os01g74450, and (**F**) Os01g06740 between Ilpum and NIL under the same conditions.

**Table 1 ijms-22-10696-t001:** Top 20 differentially expressed (DEGs) genes between Ilpum and near-isogenic line.

No.	Accession Number	MSU ID	Log2 FPKM (Val_2/Val_1)	Log2 FPKM (Val_2/Val_1)	Description	Regulatory Pathway Involved (MapMan)
			Ilpum	Near-Isogenic Line		
Genes upregulated in Ilpum but downregulated in near-isogenic line (Shingwang × Ilpum)						
1	Os03t0177900-02	LOC_Os03g08050	12.628631	−9.878618	Elongation factor 1-alpha, putative, expressed	Protein biosynthesis, translation elongation, eEF1 aminoacyl-tRNA binding factor activity
2	Os11t0106400-03	LOC_Os11g01510	11.559828	−3.195272	Ubiquitin-activating enzyme E1.	Protein homeostasis, ubiquitin-proteasome system, ubiquitin conjugation (ubiquitylation)
3	Os11t0545600-02	LOC_Os11g34300	9.600705	−0.439048	Chromatin modification-related protein EAF3, putative, expressed	Chromatin organization, histone modifications, histone lysine methylation
4	Os01t0549700-03	LOC_Os01g36890	9.582753	−4.550572	DEAD-box ATP-dependent RNA helicase, putative, expressed	Not assigned, not annotated
5	Os01t0681600-01	LOC_Os01g48930	8.685934	−0.853614	Splicing factor-related, putative, expressed	Not assigned, not annotated
7	Os01t0674500-01	LOC_Os01g48370	8.416789	−0.569737	OsFBT1-F-box and tubby domain-containing protein, expressed	Not assigned, not annotated
8	Os06t0669400-02	LOC_Os06g45820	8.19466	−0.926506	Cell division OsFtsH2 FtsH protease, homologue of AtFtsH2/8, expressed, chloroplastic	Protein homeostasis, proteolysis, metallopeptidase activities, FtsH endopeptidase activities
9	Os03t0136600-01	LOC_Os03g04380	8.025587	−0.24581	Complex 1 LYR protein family protein, LYR motif-containing protein, putative, expressed	Not assigned, not annotated
10	Os02t0471500-03	LOC_Os02g27220	7.601656	−1.046997	Protein phosphatase 2C-containing protein, putative, expressed	Protein modification, phosphorylation, serine/threonine protein phosphatase superfamily
6	Os03t0626800-01	LOC_Os03g42840	7.596287	−1.571210	Calcineurin B protein	Not assigned, not annotated
Genes downregulated in Ilpum but upregulated in near-isogenic line (Shingwang × Ilpum)						
1	Os01t0835600-02	LOC_Os01g61890	−1.265655	12.479699	AT hook family, DNA-binding, conserved site domain-containing protein	Not assigned, not annotated
2	Os04t0636900-01	LOC_Os04g54440	−0.874467	12.421764	RNA-binding region RNP-1 (RNA recognition motif) domain-containing protein	Not assigned, not annotated
3	Os03t0838100-01	LOC_Os03g62180	−8.277127	11.072724	Serine/threonine protein kinase-related domain-containing protein	Not assigned, not annotated
4	Os02t0670900-05	LOC_Os02g44980	−0.165004	9.986115	Transmembrane amino acid transporter protein, putative, expressed	Not assigned, not annotated
5	Os03t0680800-05	LOC_Os03g47740	−0.632828	9.986108	Homeodomain protein, putative, expressed	Not assigned, not annotated
6	Os08t0556000-02	LOC_Os08g44200	−1.715944	9.948021	ECT5, putative, expressed	Not assigned, not annotated
7	Os04t0476100-02	LOC_Os04g40040	−0.817003	9.790734	Copper methylamine oxidase precursor	Not assigned, not annotated
8	Os11t0546000-01	LOC_Os11g34350	−0.244259	8.799661	ATP-binding cassette sub-family E member 1, putative, expressed RNase L inhibitor-like protein.	Not assigned, not annotated
9	Os04t0566500-04	LOC_Os04g47870	−0.604165	8.537675	PINHEAD, putative, expressed, similar to Argonaute protein	Not assigned, not annotated
10	Os03t0784700-01	LOC_Os03g57120	−0.125439	8.387406	Ferredoxin NADP reductase, root isozyme, chloroplast precursor (EC 1.18.1.2) (FNR)	Not assigned, not annotated

**Table 2 ijms-22-10696-t002:** Top 10 exponentially induced genes in NIL compared to Ilpum.

No	Accession Number	MSU ID	Log2 FPKM	Log2 FPKM	Description	Regulatory Pathway Involved (MapMan)
			Ilpum	NIL		
1	Os02t0519900-04	LOC_Os02g32030	0.625140	15.416994	Elongation factor, putative, expressed	Not assigned, Not annotated
2	Os06t0686400-01	LOC_Os06g47200	0.511699	12.101766	Bifunctional inhibitor/plant lipid transfer protein/seed storage domain-containing proteinOsLTPG13|OsLTPg13-protease inhibitor/seed storage/LTP family protein precursor, expressed	Not assigned, Not annotated
3	Os01t0837300-01	LOC_Os01g62020	0.447945	11.544435	NAD-dependent epimerase/dehydratase family domain-containing protein, expressed, similar to UDP-glucuronic acid decarboxylase 1	Not assigned, Not annotated
4	Os01t0139200-03	LOC_Os01g04650	0.709687	10.877399	PB1 domain-containing protein, expressed	Not assigned, Not annotated
5	Os03t0300900-03	LOC_Os03g18890	1.568129	10.049414	Glycosyl transferase 8 domain-containing protein, putative, expressed	Not assigned, Not annotated
6	Os07t0583600-03	LOC_Os07g39470	2.704248	9.282445	Chitin-inducible gibberellin-responsive protein 2, gibberellin response modulator protein, putative, expressed	Not assigned, Not annotated
7	Os06t0132100-01	LOC_Os06g04130	0.165318	9.123196	Transmembrane receptor, eukaryota domain-containing protein. Lung seven transmembrane domain-containing protein, putative, expressed	Not assigned, Not annotated
8	Os02t0490500-01	LOC_Os02g28900	0.838091	8.843985	Cytokinin-*O*-glucosyltransferase 2, putative, expressed, UDP-glucuronosyl/UDP-glucosyltransferase family protein	Not assigned, Not annotated
9	Os02t0668100-02	LOC_Os02g44780	0.817151	8.678553	Geranylgeranyl-diphosphate synthase, polyprenyl synthetase, putative, expressed	Not assigned, Not annotated
10	Os08t0560900-03	LOC_Os08g44680	1.372722	8.112715	Photosystem I reaction center subunit II, chloroplast precursor, putative, expressed	Photosynthesis, photophosphorylation, photosystem

**Table 3 ijms-22-10696-t003:** EPG information from *N. cincticeps* nymphs feeding on rice seedlings.

EPG Waveform	Frequency (Hz)	Feeding Activity
Nc1	-	None probing
Nc2	Irregular	Penetration
Nc3	Irregular	Path and salivation
Nc4	3.5–5.6	Xylem feeding [82]
Nc5	0.01–0.2	Salivation or physical activity in phloem [82]
Nc6	2.6–6.2	Phloem sap sucking [79]

## Data Availability

The data presented in this study are available in the Appendix A. In addition, we submitted the gene expression data (Processed data after MapMan analysis) and raw sequence data (RNA-Seq raw data) to the Gene Expression Omnibus (GEO) and Short Read Archive (SRA) at NCBI (https://www.ncbi.nlm.nih.gov/geo/submitter/), accessed on 25 August 2021, with accession numbers GSE176497 and SRP323419, respectively.

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
