# Peer review of "RNA-Seq and Electrical Penetration Graph Revealed the Role of Grh1-Mediated Activation of Defense Mechanisms towards Green Rice Leafhopper (Nephotettix cincticeps Uhler) Resistance in Rice (Oryza sativa L.)"

_ijms, 2021, doi:10.3390/ijms221910696_

Round 1
Reviewer 1 Report
Thanks to the authors for the comments and corrections.
I accept the manuscript in the form provided by the authors
Author Response
We are thankful to the reviewer for his valuable comments, which helped us improve substantially the manuscript.
Reviewer 2 Report
I can confirm that the subject matter of this study (RNA-Seq and Electrical Penetration Graph Revealed the Role of 2 Grh1-Mediated Activation of Defense Mechanisms towards 3 Green Rice Leafhopper (Nephotettix cincticeps Uhler) Resistance 4 in Rice (Oryza sativa L.) is of interest and relevance for publication in International Journal of Molecular Sciences
Comments to the Authors:
- The authors did not present any hypothesis at the beginning
- correct Conclusion - add a practical implications statement. In my opinion this section may be improved reducing data and giving few key message/take home message to the readers. An idea may be to synthetize in 3-5 bullet the key results of the study, evidences and recommendation. This improvement will increase clearness and readability. Add a practical implications statement
- Extensive English editing and reforming is highly recommended
Author Response
RNA-Seq and Electrical Penetration Graph Revealed the Role of Grh1-Mediated Activation of Defense Mechanisms towards Green Rice Leafhopper (Nephotettix cincticeps Uhler) Resistance in Rice (Oryza sativa L.)
Manuscript ID ijms-1375403
Point by point reply to the comments of reviewer
|
Reviewer 2 |
|
|
I can confirm that the subject matter of this study (RNA-Seq and Electrical Penetration Graph Revealed the Role of 2 Grh1-Mediated Activation of Defense Mechanisms towards 3 Green Rice Leafhopper (Nephotettix cincticeps Uhler) Resistance 4 in Rice (Oryza sativa L.) is of interest and relevance for publication in International Journal of Molecular Sciences |
We would like to thank the reviewer for his valuable comments to improve our manuscript. We appreciate the encouraging comments and the interest shown in our manuscript. We have tried to improve the manuscript following the reviewer’s comments.
|
|
Comments to the Authors: |
|
|
The authors did not present any hypothesis at the beginning |
We appreciate the concern raised by the reviewer. We have included a statement of purpose and hypothesis in the introduction section as suggested. See lines 125–131, as follows: “This study aimed at investigating the changes that would occur at a transcriptome-wide level, and the defense mechanisms and signaling pathways activated when rice plants are exposed to green rice leafhopper (GRH) infestation. To achieve that, we used rice near isogenic lines (NILs) reported by Park et al. [33] (carrying Green rice leafhopper resistance 1 (Grh1) locus, which has been fine mapped to chromosome 5 and located in a region covered by 670 kbp), and the rice cultivar Ilpum (GRH-susceptible) exposed to GRH infestation. Then, we performed a transcriptome study to identify novel GRH-responsive genes between the NIL-GRH resistant line and Ilpum.” |
|
Correct Conclusion - add a practical implications statement. In my opinion, this section may be improved reducing data and giving few key message/take home message to the readers. An idea may be to synthetize in 3-5 bullet the key results of the study, evidences and recommendation. This improvement will increase clearness and readability. Add a practical implications statement Extensive English editing and reforming is highly recommended
|
We appreciate the suggestion made by the reviewer to improve the conclusions. Therefore, we have made necessary changes in the conclusions as suggested (lines 899–913).
|

This manuscript is a resubmission of an earlier submission. The following is a list of the peer review reports and author responses from that submission.
Round 1
Reviewer 1 Report
Overall the article is good, but I have a few comments:
1 – Is rice the only cereal crop cultivated for human consumption?
2 –Section: “From a general perspective, insects are said to be key components of life on our planet, accounting for over 50% of known organisms [4], and their presence has proven essential for maintaining balanced terrestrial ecosystems. In the same way, insects have been shown to play fundamental roles in food production, and their absence would com-promise it [5,6]. In contrast, humans and insects have been constantly competing for food and fiber, while insects are said to be responsible for yield and economic losses annually to food crops and other plants across the world [7,8]. Scientists have proposed various measures to control and minimize these losses, and the most practical and economical control measure is the use of resistant varieties to insect pests.”
General information about the significance of insects is not necessary to mention in the text, focus only on the significance of your research.
3- In the sentence: «Concerning the cellular respiration regulation genes, 32.6% and 67.4% were found to be down- and up-regulated in Ilpum, respectively, against 47.4% and 52.6% (down- and up-regulated, respectively) in NIL. In contrast, the majority of the genes involved in carbohydrate metabolism and expressed in response to GRH were down-regulated in Ilpum (61.6% against 38.4% up-regulated),..»
check the correctness of the brackets in the underlined fragment.
Author Response
Green Rice Leafhopper (Nephotettix cincticeps Uhler)-Mediated Transcriptome Triggered Multiple Regulatory Pathways in Rice (Oryza sativa L.)
Manuscript ID: ijms-1280159
Point by point response to the comments of reviewers
We are thankful to the editorial team and anonymous reviewers for their time given to this manuscript. We appreciate their comments, and are happy to share that most of the comments are addressed and have substantially improved the quality of the manuscript. We would like to specify that all changes in the manuscript were highlighted green, no track change was applied to the manuscript. We hope that the manuscript in the present form will be suitable for publication in the journal.
|
Reviewer 1 |
|
|
Overall the article is good, but I have a few comments: |
We are thankful to the reviewer for his valuable comments and observations that helped us substantially improve the manuscript. We have tried to address the question raised by the reviewer to the best our understanding. |
|
1 – Is rice the only cereal crop cultivated for human consumption? |
Here, we intended to say that rice is the only cereal crop solely cultivated for human consumption. We would like to apologize for the inconvenience caused by the word “solely”. |
|
2 –Section: “From a general perspective, insects are said to be key components of life on our planet, accounting for over 50% of known organisms [4], and their presence has proven essential for maintaining balanced terrestrial ecosystems. In the same way, insects have been shown to play fundamental roles in food production, and their absence would com-promise it [5,6]. In contrast, humans and insects have been constantly competing for food and fiber, while insects are said to be responsible for yield and economic losses annually to food crops and other plants across the world [7,8]. Scientists have proposed various measures to control and minimize these losses, and the most practical and economical control measure is the use of resistant varieties to insect pests.” |
Were have rephrased the paragraphs of the introduction as suggested by the reviewers. |
|
General information about the significance of insects is not necessary to mention in the text, focus only on the significance of your research. |
|
|
3- In the sentence: «Concerning the cellular respiration regulation genes, 32.6% and 67.4% were found to be down- and up-regulated in Ilpum, respectively, against 47.4% and 52.6% (down- and up-regulated, respectively) in NIL. In contrast, the majority of the genes involved in carbohydrate metabolism and expressed in response to GRH were down-regulated in Ilpum (61.6% against 38.4% up-regulated)»
check the correctness of the brackets in the underlined fragment. |
We are thankful to the reviewer for his observations. We have revised this section in the discussion accordingly:
|

Reviewer 2 Report
This manuscript presents a transcriptome analysis of rice cultivars exposed to Green rice leafhopper (GRH) feeding. The comparison between the plant’s genetic level response with a quantitative measurement of the pests feeding (Electrical Penetration Graph tests (EPG)) makes this manuscript very unique. Findings indicate a systematic response to feeding that includes the activation of genes related to defense, pathogen attack, but also abiotic stress, metabolism and housekeeping genes. The manuscript relates these findings to plant defenses in the context of food security.
Unfortunately, the manuscript suffers from several deficiencies that limit the impact of its findings. Particularly, the integration of the EPG data and the transcriptome data is extremely limited. In addition, the methods and results lack sufficient information to evaluate crucial aspects of the data. As a result, the manuscript appears to be a list of genes that are down- or up- regulated across cultivars without much interpretation or integration into a broader context of plant defenses. Below I provide specific comments in hopes to contribute to the improvement of this manuscript.
Abstract
The abstract should be revised to emphasize the findings and avoid repeating details from the methods or using sentences that are vague and uninformative.
Here are some examples:
- The use of qPCR to validate data could be removed from the abstract, or its results better emphasized, as opposed to stating the method.
- The use of the words ‘interestingly’, ‘interesting’ should be avoided, and instead the authors should indicate the findings that are relevant. There is nothing wrong with the word itself, but the current form of the abstract uses it several times in otherwise vague sentences without much information about the findings.
- ‘which prove useful in understanding molecular mechanisms associated with GRH induced biotic stress responses’. How are these useful? How do we understand these mechanisms better after learning about your findings? This should be explicitly stated here.
- What are your central findings and how do they inform the broader context of plant defenses?
General comments about the text.
Text needs to be revised for clarity and specificity. There are several instances with missing information, incomplete sencenes and odd changes in formatting. There are other instances where the text would benefit from more clarity to strengthen the document.
Here are some selected examples:
- methods paragraph 1 – ‘…enriched soil (company name)…’ – what is the company name? Even more importantly: what is the composition of this soil?
- RNAseq paragraph 2: ‘The quality control…’ – The first sentence of the paragraph is missing the verb and thus not a full sentence. Please revise.
- ‘male GRH inoculated…’. I believe inoculated is typically used for microbe addition. I suggest the authors consider using a term that is traditionally used for the addition of insects.
- ‘exclude fall down insects…’. Insects that fell down? Clarify if it is some other specific behavior.
- last paragraph of section 2.1 seems to have different font. The last sentence is also unclear: ‘In the same perspective, we pooled the top 10 exponentially induced genes in NIL (…) while showing lower expression levels in Ilpum under the same conditions…’. What do you mean by pooled here? What perspective? I am having trouble figuring out how this describes the results.
- ‘… to uptake phloem suck, the process of wave formation…’ this first part of the sentence needs revising.
Introduction:
The introduction presents information that seems disorganized and, at times, redundant. I believe the manuscript would greatly benefit from streamlining the introduction to emphasize the uniqueness of this dataset.
Here are some selected observations:
- The two first paragraphs seem to be the same topic.
- Redundancy should be avoided. If absolutely necessary to revisit a topic, it should be clear how information is added each time. For example, GRH is presented as vector of diseases in paragraph 3, and 4, with no evident benefit of the redundancy and no additional detail is provided.
- reference to Matsumoto and Hattori seems the reasoning for this study, but it is poorly connected to the firts part of the intro, and poorly explained. What were the enzymes that were detected? Could this help explain at least part of the results in this manuscript? It seems like the connection could be more specific in a way that strengthens this work. Perhaps the authors can use those studies to provide expected results and interpret the data in its biological context.
- Avoid vague statements and instead provide specific information. For example, the paragraph on plant genetic response to herbivory provides only very vague information. What do you mean with a ‘myriad’? how many?, ‘other cellular processes’ which ones? Perhaps because of this lack of information, this paragraph seems out of place.
- The RNAseq paragraph is also very vague. Specifically address how this technology and associated methods can help resolve issues with plant defenses. This paragraph also lacks any citations. The authors should provide sufficient and specific context and references.
- Information on EPG test at the end of the introduction seems out of place. I suggest the authors include this information in the methods and instead find a place in the introduction to explicitly state how combining EPG data with transcriptomics can inform our understanding of plant defenses. I agree with the authors that this is a unique approach, but the impact needs to be clearer in the introduction.
- Importantly: The Objective of comparing transcriptomes between GRH resistant and non resistant rice cultivars’ is clear, but not new, and the conceptual link between this comparison and the EPG test is unclear.
Methods:
The methods must be much clearer and provide details in an organized fashion for better evaluation of their fit with the proposed study.
Some examples of missing details that are key to evaluating the study:
- What was the sample size used for the pest induction experiments? How many seeds did you use? How many seedlings? How many of these were processed for transcriptomics?
- Were seedlings processed to assess non-pest induced transcriptomics? It seems they were from the results, but the methods are not clear as to how many or how they were handled. Were they grown in the same conditions as the ones treated with GRH?
- How many cultivars were used? Methods indicate 2 but Table 3 shows 3
- What was the origin and composition of the enriched soil?
- How did you define low quality reads for removal?
- From fig5, it looks like survival rates were recorded. How was this established? What was the sample size? How was it analyzed? Which plants were used? Were these the same plants used for transcriptomes and/or for EPG tests?
Some examples of information that is provided in disorder. Please aim to present the methods in the order they are used or appear in your methods. Your reader may feel lost as the details are not presented in a logical order.
- what was the RNA extraction kit or protocol?
- what was the RNA/DNA Purification protocol? The protocol for cDNA transcription? This should be presented in order.
- What were the trimming methods used? Please specify software and parameters.
- Where are the methods for qPCR validation of RNA seq results?
Some of the details missing raise concerns about the validity of the study and the authors should take great care to explain. Here are the most critical issues I encountered:
- Did the authors sequence uninfected controls? These are mentioned in the results but not accounted for in the methods. How were these treated? What conditions where they grown? How did the authors ensure they were otherwise identical to the seedlings treated with GRH?
- What was the sample size? It is important to know how many plants were processed and whether their transcriptome data was pooled. From figure 1D, it looks like there is more than one transcriptome per cultivar, does this represent 2 plants per cultivar? How do the authors explain the differences?
- ‘There is no study to analyze transcriptome of rice which has Grh1 …’ This sentence seems extremely important in highlighting one of the most unique features of this paper: the fact that you are comparing transcriptome to EPG tests. Yet it seems out of place in this paragraph in the methods and it is difficult to understand. The authors should consider moving this sentence to the last paragraph of the introduction to highlight its uniqueness and revise the sentence for clarity.
The use of EPG test technology in this context seems like one of the most important aspects of this paper, yet the EPG test method is poorly explained.
- How did the authors classify the waveforms into groupings Nc1-Nc7? The EPG literature typically provides specific measurements that include amplitude, repetition rate, time to penetration, duration of penetration, duration of each pattern, number of sequences, etc.
- The authors refer to the waveform classification used in Kimmins 1989, yet the pest species in that study is Nilaparvata lugens a different species of leafhopper. Therefore, much more information on the waveform patterns needs to be recorded and analyzed before directly comparing the classification system. If the authors already did this, this proof of the classification should be explicit in the methods and presented in the results.
- how does EPG test alter grasshopper feeding behavior?
- why were only males used for this test?
- where the same plants used for transcriptome analysis and for the EPG test on grasshoppers? This would allow the authors to directly compare transcriptome data to the insect’s sucking activity, perhaps explaining the variability in transcriptome patterns showing in fig1D?
- was this test done in Shingwang cultivar too? The results table seems to suggest it was, but it is not mentioned in the methods.
Results:
In general, I suggest the authors present some of this data in form of tables. The figures are helpful in establishing the findings for the transcriptome, but there is no indication of results from EPG tests. Because of lack of detail in the methods it is also difficult to interpret some of these results. Here I provide few select comments that may improve the results section:
- ‘major characteristic of NIL…’ this information on the genetic lines must be presented earlier, perhaps in the intro, as reasoning to choose it for analysis or consideration for interpreting data.
- results on EPG on page 16 should have a subtitle.
- ‘…high and sustained GRH survival percentage over time…’ sounds like the grasshopper survived. Do the authors mean the grasshopper or the plant survival? If it is the grasshopper, why is its survival low in non-resistant lines? The collection of survival data is absent from the methods.
- Authors should provide clear indication of the waveforms results other than those provided in Table 3.
Figure 1c: how is there 0 in all intersections between up regulated and down regulated? It seems only differentially recgulated genes were used. But unclear if comparisons are within cultivar or species. This makes me question whether this venn diagram is the best choice for this data.
Fig 1D: variability – it seems it represents at least 4 samples (should be explained somewhere). Interestingly, Ilpum seems to vary in upregulation. What does this mean? How general are these patterns?
Fig 5: It seems there was some confusion in the caption or legend of this figure. The legend indicates the red color and black data points represent Ilpum, yet the caption indicates the opposite, that it represents NIL. Although it seems the caption provides with the methods behind this figure, these methods are incomplete (what is the sample size?), and should be included in the methods section, not in the figure caption. Are the error bars representing standard error of the mean? How was the percent survival calculated? Are these grasshoppers on separate plants? Or in groups?
Table 3: Information in this table could be further clarified. As I mentioned before, the authors should first explain how the Nc1-7 classification was obtained, thus clearly providing with the origin of this table’s data. Are the numeric values all minutes? It looks like the authors are providing an average and standard deviation or standard error of the mean. Please specify which one. Because the methods are incomplete, it is unclear how many samples are being averaged here, and thus the meaning of variance around the mean is difficult to establish. The Shingwang cultivar is included here but was not mentioned in the methods. Where these grown in the same way? The table caption could be more informative as to some of these details.
Discussion:
The discussion could be strengthened by better fitting the findings in the context of plant defenses. Because of issues I have detailed in the introduction, methods and results, it is difficult to provide more specific advice. I believe once those issues are resolved, it will be much easier to streamline the discussion and focus on how this unique dataset contributes to our understanding of plant-insect interactions and its consequences for food security. Here, I provide few comments to the current version of the discussion:
- first paragraph is a vague, and generalized summary on plant defenses. I suggest authors identify the important information and highlight with specific information that relates to their study’s findings or remove this paragraph all together.
- ‘… GRH-infestation activated typical basal defense system similar to that observed during pathogen attacks’. Which pathogens are you referring to? Include citations.
- The authors indicate that redox homeostasis is higher in NIL than Ilpum, what implications can this have for plant defenses and regulatory mechanisms?
- I suggest the authors refrain from summarizing the results in the discussion and instead provide interpretation of the patterns by placing it in the broader context.
- The authors conclude that pest resistance is mediated by transcriptome wide interactions, instead of gene-gene action. They do highlight the variety of pathways that are differentially regulated in resistant vs non-resistant cultivars, yet I am failing to see how these findings indicate a network of genome wide interactions. Perhaps better framing on how the regulatory mechanisms connect, would support this conclusion? Also, I am not certain this, by itself is a novel finding. The authors should better place this finding in the context of appropriate literature for the reader to appreciate its novelty.
- Unfortunately, the EPG test data are very poorly integrated in the discussion. It is unclear how/if they contribute any information to the interpretation of transcriptome data. I believe this could give the discussion a unique dimension.
Author Response
Green Rice Leafhopper (Nephotettix cincticeps Uhler)-Mediated Transcriptome Triggered Multiple Regulatory Pathways in Rice (Oryza sativa L.)
Manuscript ID: ijms-1280159
Point by point response to the comments of reviewers
We are thankful to the editorial team and anonymous reviewers for their time given to this manuscript. We appreciate their comments, and are happy to share that most of the comments are addressed and have substantially improved the quality of the manuscript. We would like to specify that all changes in the manuscript were highlighted green. We hope that the manuscript in the present form will be suitable for publication in the journal.
|
Reviewer 2 |
|
|
This manuscript presents a transcriptome analysis of rice cultivars exposed to Green rice leafhopper (GRH) feeding. The comparison between the plant’s genetic level response with a quantitative measurement of the pests feeding (Electrical Penetration Graph tests (EPG)) makes this manuscript very unique. Findings indicate a systematic response to feeding that includes the activation of genes related to defense, pathogen attack, but also abiotic stress, metabolism and housekeeping genes. The manuscript relates these findings to plant defenses in the context of food security. Unfortunately, the manuscript suffers from several deficiencies that limit the impact of its findings. |
We would like to thank the reviewer for the time taken to analyze our manuscript with scrutiny, and for the valuable observations and comments made to improve the quality of the manuscript as well as the significance of the results. We would like to say that almost all the comments have been addressed, and the current version of the manuscript has been substantially revised accordingly.
|
|
Particularly, the integration of the EPG data and the transcriptome data is extremely limited. |
|
|
In addition, the methods and results lack sufficient information to evaluate crucial aspects of the data. As a result, the manuscript appears to be a list of genes that are down- or up- regulated across cultivars without much interpretation or integration into a broader context of plant defenses. Below I provide specific comments in hopes to contribute to the improvement of this manuscript. |
We appreciate the reviewer’s comment, and have tried to describe the results with details in order to provide additional information for a better understanding of the methods used to perform the experiments by readers (see lines 1705–1879) |
|
Abstract |
|
|
The abstract should be revised to emphasize the findings and avoid repeating details from the methods or using sentences that are vague and uninformative. Here are some examples: |
We are thankful to the reviewer for his valuable comments. We have improved the abstract focusing on significance of the findings as suggested, after removing what was identified as unnecessary description (Lines 15–176) |
|
- The use of qPCR to validate data could be removed from the abstract, or its results better emphasized, as opposed to stating the method. |
We have removed this part from the abstract as suggested. |
|
- The use of the words ‘interestingly’, ‘interesting’ should be avoided, and instead the authors should indicate the findings that are relevant. There is nothing wrong with the word itself, but the current form of the abstract uses it several times in otherwise vague sentences without much information about the findings. |
We have deleted the word interestingly from the abstract as suggested.
|
|
- ‘which prove useful in understanding molecular mechanisms associated with GRH induced biotic stress responses’. How are these useful? How do we understand these mechanisms better after learning about your findings? This should be explicitly stated here. - What are your central findings and how do they inform the broader context of plant defenses? |
We have rephrased this statement in the abstract as suggested by the reviewer. Addition information have been provided for clear understanding of the findings. |
|
General comments about the text: |
|
|
Text needs to be revised for clarity and specificity. There are several instances with missing information, incomplete sentences and odd changes in formatting. There are other instances where the text would benefit from more clarity to strengthen the document. |
We are thankful to the reviewer for his constructive comments tending to improve the quality of our manuscript. We have revised thoroughly the manuscript, and we have tried to specify or improve confusing statements, while incorporating the suggestions. |
|
Here are some selected examples: |
|
|
· methods paragraph 1 – ‘…enriched soil (company name)…’ – what is the company name? Even more importantly: what is the composition of this soil? · · · · · RNAseq paragraph 2: ‘The quality control…’ – The first sentence of the paragraph is missing the verb and thus not a full sentence. Please revise. · ‘male GRH inoculated…’. I believe inoculated is typically used for microbe addition. I suggest the authors consider using a term that is traditionally used for the addition of insects. · ‘exclude fall down insects…’. Insects that fell down? Clarify if it is some other specific behavior. · · last paragraph of section 2.1 seems to have different font. · · The last sentence is also unclear: ‘In the same perspective, we pooled the top 10 exponentially induced genes in NIL (…) while showing lower expression levels in Ilpum under the same conditions…’. What do you mean by pooled here? What perspective? I am having trouble figuring out how this describes the results. · ‘… to uptake phloem suck, the process of wave formation…’ this first part of the sentence needs revising. |
We have added the company name (line 726–803). However, the company due to evident reasons did not disclose this soil composition, we assume. The soil bags do not contain any specification in this regards. This is similar to the Hoagland salt and sodium nitrate used to prepare nutrient solution for hydroponic culture (the composition of the medium has never been disclosed by the manufacturer).
We have added the verb as suggested (line 871).
We apologize for the inconvenience. We have replaced “GRH inoculated” with “GRH infested (Line 1808).
We have remove have removed this statement to avoid confusing interpretation by the readers.
We thank the reviewer for his vigilance. We have harmonized the font in the pointed paragraph with the rest of the manuscript (Palatino Linotype) Line 741–742: we have rephrased as followed: In the same way, the transcript accumulation of the top 10 exponentially induced genes indicated up-regulated by 8.1–15.4 log2FC in NIL, while showing lower expression levels in Ilpum under the same conditions (up-regulated by 0.2–2.7 log2FC) (Table 2). We apologize for the inconvenience. We have revised this statement as suggested. |
|
Introduction: The introduction presents information that seems disorganized and, at times, redundant. I believe the manuscript would greatly benefit from streamlining the introduction to emphasize the uniqueness of this dataset. |
|
|
· Here are some selected observations: · The two first paragraphs seem to be the same topic. Redundancy should be avoided. If absolutely necessary to revisit a topic, it should be clear how information is added each time. For example, GRH is presented as vector of diseases in paragraph 3, and 4, with no evident benefit of the redundancy and no additional detail is provided. |
We would like to thank the reviewer for his positive suggestions. We have revised the introduction section as suggested (Lines 182–191; 198–209; 493–499).
|
|
· reference to Matsumoto and Hattori seems the reasoning for this study, but it is poorly connected to the first part of the intro, and poorly explained. What were the enzymes that were detected? Could this help explain at least part of the results in this manuscript? It seems like the connection could be more specific in a way that strengthens this work. Perhaps the authors can use those studies to provide expected results and interpret the data in its biological context. |
We are thankful to the reviewer for his raised concerns. In the introduction we have revised the paragraph in lines 493–499. In addition, we have removed from the introduction the statement referring to Hattori. |
|
· Avoid vague statements and instead provide specific information. For example, the paragraph on plant genetic response to herbivory provides only very vague information. What do you mean with a ‘myriad’? how many?, ‘other cellular processes’ which ones? Perhaps because of this lack of information, this paragraph seems out of place. |
We appreciate the concern raised by the reviewer, and we have improved this section accordingly. We believe that at this section, one cannot specify the number of genes with accuracy. This is the reason why use words such as myriad, several, many, or a number of …, which to our knowledge are not wrong to be employed in the introduction. |
|
· The RNAseq paragraph is also very vague. Specifically address how this technology and associated methods can help resolve issues with plant defenses. This paragraph also lacks any citations. The authors should provide sufficient and specific context and references. |
We would like to thank the reviewer for the concerns raised. We have added a couple of references with regard to the use of RNA-seq technology in plant defense. |
|
· Information on EPG test at the end of the introduction seems out of place. I suggest the authors include this information in the methods and instead find a place in the introduction to explicitly state how combining EPG data with transcriptomics can inform our understanding of plant defenses. I agree with the authors that this is a unique approach, but the impact needs to be clearer in the introduction. · Importantly: The Objective of comparing transcriptomes between GRH resistant and non resistant rice cultivars’ is clear, but not new, and the conceptual link between this comparison and the EPG test is unclear. |
We have revised the last paragraph of the introduction section as suggested, and have moved the paragraph in question to the materials and methods. |
|
Methods: The methods must be much clearer and provide details in an organized fashion for better evaluation of their fit with the proposed study. |
|
|
Some examples of missing details that are key to evaluating the study: - What was the sample size used for the pest induction experiments? How many seeds did you use? How many seedlings? How many of these were processed for transcriptomics?
- Were seedlings processed to assess non-pest induced transcriptomics? It seems they were from the results, but the methods are not clear as to how many or how they were handled. Were they grown in the same conditions as the ones treated with GRH? - How many cultivars were used? Methods indicate 2 but Table 3 shows 3
- What was the origin and composition of the enriched soil?
- How did you define low quality reads for removal?
- From fig5, it looks like survival rates were recorded.
How was this established? What was the sample size? How was it analyzed?
Which plants were used? Were these the same plants used for transcriptomes and/or for EPG tests? |
We appreciated the concern raised by the reviewer. We would like to specify that concerning the number seeds for germination, we have sown about 50 seedlings in a 50-well tray, but only seedlings with uniform height were used for the experiment as specified in lines 1728–1815).
We thank the reviewer for the concern. We would like to specify that leaf samples for RNA-seq were collected from two different plants in duplicate for each treatment (control seedlings with no GRH and GRH infested seedlings) as indicated in line 1815.
Two rice lines were used to perform RNA-Seq analysis. As indicated in lines 1720–1722, the Near Isogenic Line (NIL) was developed from a cross between Shingwang (Grh1 donor parent) and Ilpum (recurrent parent). For this reason, Shingwang was included in the study only during Electrical penetration graph (EPG) test to investigate the feeding behavior of GRH between Ilpum (susceptible) and NIL (harboring Grh1) as compared to Shingwang.
We would like to indicate that the manufacturer due to evident reasons could not disclose the information on the chemical composition of the rice soil used in the experiment during seedling growth. Therefore, we are not able to provide this information. A similar case was found for Hoagland salt and calcium nitrate used to prepare Hoagland solution, the manufacturer does not disclose the chemical composition of the medium.
We thank the reviewer for his concern. We would like to specify that this task was done by the sequencing company (Macrogen, Daejeon) following standard protocol as indicated in lines 1878–1886.
We sincerely apologize for the inconvenience. We have discover a labelling problem in this figure (previously figure 5). The high survival rate of GRH is for Ilpum and the lower survival rate is for NIL. We have corrected the figure label and in lines and the description accordingly (lines 513–571. Lines 1838–1841: the survival rate of GRH in Ilpum and NIL GRH-infested seedlings was determined as follows: [(total number alive insects (GRH) initially infestednumber of dead insects over time)]/(total number of alive insects initially infested)100.
We would like to indicate that it is not practically convenient to collect samples from seedlings during EPG test to avoid bias on the readings. For this reason, another set of the same genetic materials grown under the same conditions were infested with GRH for RNA-Seq analysis. |
|
Some examples of information that is provided in disorder. Please aim to present the methods in the order they are used or appear in your methods. Your reader may feel lost as the details are not presented in a logical order. - what was the RNA extraction kit or protocol?
- what was the RNA/DNA Purification protocol? The protocol for cDNA transcription? This should be presented in order.
- What were the trimming methods used? Please specify software and parameters. - Where are the methods for qPCR validation of RNA seq results? |
We appreciate the concerns raised by the reviewer. We have provided the required information in lines 1843–1844.
As indicated in lines 986, trimming was performed using Trimmonmatic program, see reference [81] (lines 1891–1895). The information about qPCR validation of RNA-Seq results is already available in lines 1983–2088 (under section 4.10). |
|
Some of the details missing raise concerns about the validity of the study and the authors should take great care to explain. Here are the most critical issues I encountered: - Did the authors sequence uninfected controls? These are mentioned in the results but not accounted for in the methods. How were these treated? What conditions where they grown? How did the authors ensure they were otherwise identical to the seedlings treated with GRH? |
The authors apologize for the inconvenience. We have specified that samples were collected from both control (non GRH-infested) and GRH-infested seedlings. With this precision, there is no doubt that control seedlings were used as reference. Besides, all transcript accumulation values are expressed as log2FC, which implies that FPKM2/FPKM1 (treated/control) were used and the logarithm base 2 was applied. For more information, kindly seed the public repository where the raw data and processed data were submitted (NCBI) (lines 373–379): Gene expression and raw sequence data have been submitted to the Gene Expression Omnibus (GEO) and Short Read Archive (SRA) at NCBI (https://www.ncbi.nlm.nih.gov/geo/submitter/) with accession numbers GSE176497 and SRP323419, respectively, using FileZilla 3.54.1 software (Tim Kosse, FileZilla ©2004-20021, https://filezilla-project.org/).
In addition, Plant growth conditions described how seedlings where grown. Prior to treatment (infestation), all seedlings were grown in the same way under the same conditions. Seedlings with a uniform height were selected for further steps. We did not mentioned that the control seedlings were grown separately from the GRH infested one. |
|
- What was the sample size? It is important to know how many plants were processed and whether their transcriptome data was pooled. From figure 1D, it looks like there is more than one transcriptome per cultivar, does this represent 2 plants per cultivar? How do the authors explain the differences? |
We have specified the sample size as requested. In the same section (materials and methods), we have also indicated that leaf samples in duplicate for control and GRH infested plants were sent for RNA-Seq. These samples are from two different seedlings. The results in Figure 1D show replication data (in log2FC calculated from treated and control seedlings) of the transcriptome in Ilpum and NIL. The heat map show a color scheme indicating the level of expression. It could go from near 0.1 to 4 or –4 to. Data around 0 have mixed color scheme. Grouping is done in R software by clustering of data with similar transcript patterns. |
|
- ‘There is no study to analyze transcriptome of rice which has Grh1 …’ This sentence seems extremely important in highlighting one of the most unique features of this paper: the fact that you are comparing transcriptome to EPG tests. Yet it seems out of place in this paragraph in the methods and it is difficult to understand. The authors should consider moving this sentence to the last paragraph of the introduction to highlight its uniqueness and revise the sentence for clarity. |
We apologize for the inconvenience. We have removed this statement from the manuscript. In addition, we have tempted to discuss EPG results associated with the recorded transcriptome in both Ilpum and NIL as suggested. |
|
The use of EPG test technology in this context seems like one of the most important aspects of this paper, yet the EPG test method is poorly explained. - How did the authors classify the waveforms into groupings Nc1-Nc7? The EPG literature typically provides specific measurements that include amplitude, repetition rate, time to penetration, duration of penetration, duration of each pattern, number of sequences, etc. |
We agree with the reviewer’s opinion. We have tried to discuss a bit more of this aspect in the manuscript.
In the initial version of this manuscript, the figure of waveforms was not included, and we apologize for the inconvenience. We have included the EPG results as waveforms in Figure 1. |
|
- The authors refer to the waveform classification used in Kimmins 1989, yet the pest species in that study is Nilaparvata lugens a different species of leafhopper. Therefore, much more information on the waveform patterns needs to be recorded and analyzed before directly comparing the classification system. If the authors already did this, this proof of the classification should be explicit in the methods and presented in the results. |
We have added another reference, and the classification of GRH feeding behavior Nc1 to Nc6 has been explained. |
|
- how does EPG test alter grasshopper feeding behavior?
- why were only males used for this test?
- where the same plants used for transcriptome analysis and for the EPG test on grasshoppers? This would allow the authors to directly compare transcriptome data to the insect’s sucking activity, perhaps explaining the variability in transcriptome patterns showing in fig1D?
- was this test done in Shingwang cultivar too? The results table seems to suggest it was, but it is not mentioned in the methods. |
We are not sure of understanding the essence of the concern. However, what we know is that EPG test is used to study the feeding behavior of the insect, in this case GRH (Green Rice Leafhopper). So, how it would alter, we believe that the information we get is telling how the insect feeding activity may induce changes in transcriptome profile within the plant. Regarding the use of male insects instead of female, we could say that studies on insect-plant interaction do not specify insect gender to assess the feeding behavior on crop plants. In our study, the gender was not selected with specific target, but insect were used as received from the entomology division of the Rural Administration in Korea.
As indicated earlier, it is not convenience to collect samples during EPG test. So, we used seedlings of same growth stage, but different individuals. Transcript accumulation are sensitive and may vary if seedlings exposed to a particular stress are not handled properly.
No, shingwang is the donor parent of Grh1 and was only used to develop NIL but not for the transcriptome and the recurrent parent Ilpum. We included Shingwang for comparing the feeding behavior of GRH in susceptible and resistant line NIL harboring Grh1. |
|
Results: In general, I suggest the authors present some of this data in form of tables. The figures are helpful in establishing the findings for the transcriptome, but there is no indication of results from EPG tests. Because of lack of detail in the methods it is also difficult to interpret some of these results. Here I provide few select comments that may improve the results section: |
We appreciate the suggestion made by the reviewer. However, we have summarized in Table S3 the transcriptome data between Ilpum and NIL. For the rest of data, it was convenient to present in Figures each category of regulatory pathway and metabolic process activated upon GRH infestation into rice plants for clear visualization. Nevertheless, raw data and processed dataset has been submitted to the public repository and can be freely accessed following below accession numbers (lines 375-379: Gene expression and raw sequence data have been submitted to the Gene Expression Omnibus (GEO) and Short Read Archive (SRA) at NCBI (https://www.ncbi.nlm.nih.gov/geo/submitter/) with accession numbers GSE176497 and SRP323419, respectively, using FileZilla 3.54.1 software (Tim Kosse, FileZilla ©2004-20021, https://filezilla-project.org/).
|
|
- ‘major characteristic of NIL…’ this information on the genetic lines must be presented earlier, perhaps in the intro, as reasoning to choose it for analysis or consideration for interpreting data. - results on EPG on page 16 should have a subtitle. - ‘…high and sustained GRH survival percentage over time…’ sounds like the grasshopper survived. Do the authors mean the grasshopper or the plant survival? If it is the grasshopper, why is its survival low in non-resistant lines? The collection of survival data is absent from the methods. - Authors should provide clear indication of the waveforms results other than those provided in Table 3. |
We are thankful to the reviewer for the valuable suggestion. We have briefly inserted a statement on why we have used NIL in this study.
We have added a substile as suggested.
We have provided the revision about this section earlier (above). We have indicated that there was a labelling switch, and we have corrected it. In addition, we have added a brief description of the estimation of the survival rate of GRH in both Ilpum and NIL conforming to the Figure 2 (previously figure 5) legend.
The waveforms results have been added as Figure 1 in support to numeric data in Table S1.
|
|
Figure 1c: how is there 0 in all intersections between up regulated and down regulated? It seems only differentially recgulated genes were used. But unclear if comparisons are within cultivar or species. This makes me question whether this venn diagram is the best choice for this data. |
We would like to put into context these results in question. Ilpum and NIL share the same genomic make up, except the Grh1 harbored by the NIL. To better under this figure 1c shows the up and down-regulated genes in Ilpum and NIL as well as commonly regulated genes that are in intersections between Ilpum and NIL (2 660 for upregulated and 2 775 for down-regulated). Therefore, it appears clearly that the comparison is done between Ilpum and NIL not within rice line. So, this Venn diagram strands, and used data expressed as Log2FC not relative expression which generally show expression values in control and treated samples separately. |
|
Fig 1D: variability – it seems it represents at least 4 samples (should be explained somewhere). Interestingly, Ilpum seems to vary in upregulation. What does this mean? How general are these patterns? |
We thank the reviewer for the concern. We have specified as earlier indicated. We see up and down regulation patterns with different intensities between the two seedlings. The color key helps clarify this. To us, values around zero are up or down, and due to the size of the transcriptome dataset, this figures help visualize the transcriptome profile. |
|
Fig 5: It seems there was some confusion in the caption or legend of this figure. The legend indicates the red color and black data points represent Ilpum, yet the caption indicates the opposite, that it represents NIL. Although it seems the caption provides with the methods behind this figure, these methods are incomplete (what is the sample size?), and should be included in the methods section, not in the figure caption.
Are the error bars representing standard error of the mean? How was the percent survival calculated? Are these grasshoppers on separate plants? Or in groups? |
We apologize for the inconvenience. In fact, as indicate earlier, there was a switch in figure labels, but the right interpretation of the data in Figure 2 (previously figure 5) is that explained in the figure caption. Thus, we have corrected and explained in lines.
We have specified in the legend of figure 2. We have indicated the formula used to estimate the survival rate of GRH in Ilpum and NIL GRH-infested seedlings.
|
|
Table 3: Information in this table could be further clarified. As I mentioned before, the authors should first explain how the Nc1-7 classification was obtained, thus clearly providing with the origin of this table’s data. Are the numeric values all minutes? It looks like the authors are providing an average and standard deviation or standard error of the mean. Please specify which one. Because the methods are incomplete, it is unclear how many samples are being averaged here, and thus the meaning of variance around the mean is difficult to establish. The Shingwang cultivar is included here but was not mentioned in the methods. Where these grown in the same way? The table caption could be more informative as to some of these details. |
We apologize for the inconvenience. As this question comes back again, we would say that explanation provided earlier that a new figure was added showing the waveforms, coupled with the existing classification of Nc and their specific meaning. For the number of samples used during EPG, all have been added to the manuscript in materials and methods. The inclusion of Shingwang in the study for EPG test has been clarified. |
|
Discussion: The discussion could be strengthened by better fitting the findings in the context of plant defenses. Because of issues I have detailed in the introduction, methods and results, it is difficult to provide more specific advice. I believe once those issues are resolved, it will be much easier to streamline the discussion and focus on how this unique dataset contributes to our understanding of plant-insect interactions and its consequences for food security. Here, I provide few comments to the current version of the discussion: |
We are thankful to the reviewer for the valuable suggestions. The manuscript has been thoroughly revised following recommendations. |
|
- first paragraph is a vague, and generalized summary on plant defenses. I suggest authors identify the important information and highlight with specific information that relates to their study’s findings or remove this paragraph all together. |
We appreciate the suggestion made by the reviewer. However, we think that in this context, putting into a broad context prior to narrow down to the findings in the study. We believe keeping this paragraph would not loose readers or inflate the manuscript. However, we have tried to modify a bit accordingly. |
|
- ‘… GRH-infestation activated typical basal defense system similar to that observed during pathogen attacks’. Which pathogens are you referring to? Include citations. |
We have improved this paragraph as indicated in the discussion section, which has been extensively revised. |
|
- The authors indicate that redox homeostasis is higher in NIL than Ilpum, what implications can this have for plant defenses and regulatory mechanisms? |
We are thankful to the reviewer for the concern. We have tried to discuss this matter. |
|
- I suggest the authors refrain from summarizing the results in the discussion and instead provide interpretation of the patterns by placing it in the broader context. |
We apologize for the inconvenience. We have improved the discussion section accordingly. |
|
- The authors conclude that pest resistance is mediated by transcriptome wide interactions, instead of gene-gene action. They do highlight the variety of pathways that are differentially regulated in resistant vs non-resistant cultivars, yet I am failing to see how these findings indicate a network of genome wide interactions. Perhaps better framing on how the regulatory mechanisms connect, would support this conclusion? Also, I am not certain this, by itself is a novel finding. The authors should better place this finding in the context of appropriate literature for the reader to appreciate its novelty. |
We appreciated the concerns raised by the reviewer. We understand the point of the reviewer, but we must say that this is the first transcriptome profile of GRH-induced biotic stress in rice. With this, we did not have many possibilities for comparing our study to a few other on GRH and plant defense mechanism at molecular level. We are strongly convince that looking the overall enhanced transcription profile in NIL carrying the resistance locus Grh1, compared to the susceptible Ilpum, this paint a portrait of an active interaction of Grh1 with defense related genes and signaling networks, that may lead to the observed resistance, therefore suggesting a Genome-wide regulation, rather than single loci action.. Besides, due to the existence of several species of GRH, we refrained from discussing the feeding behavior of other species to support our results in trying to relate to the recorded differential transcriptome profile between resistant and susceptible rice lines.
Nevertheless, we have tried to revise the discussion section, while considering the reviewers’ comments for improvement. |
|
- Unfortunately, the EPG test data are very poorly integrated in the discussion. It is unclear how/if they contribute any information to the interpretation of transcriptome data. I believe this could give the discussion a unique dimension. |
After including the waveforms data, we have made necessary changes with regard to the EPG test in the results as well as in the discussion section. Although, it is challenging to directly relate the feeding behavior with certainty, we have tried to suggest some hypothesis in this regards, in the discussion section, while avoiding too much speculations.. |
